# Generation of global 1-km daily soil moisture product from 2000 to 2020 using ensemble learning

Yufang Zhang[1], Shunlin Liang[2], Han Ma[2], Tao He[1], Qian Wang[3], Bing Li[4], Jianglei Xu[1], Guodong Zhang[1], Xiaobang Liu[1], Changhao Xiong[1]

[1]School of Remote Sensing and Information Engineering, Wuhan University, Wuhan 430079, China

[2]Department of Geography, The University of Hong Kong, Hong Kong 999077, China

[3]State Key Laboratory of Remote Sensing Science, Beijing Normal University, Beijing 100875, China

[4]Key Research Institute of Yellow River Civilization and Sustainable Development & Collaborative Innovation Center on Yellow River Civilization of Henan Province, Henan University, Kaifeng 475001, China

*Correspondence to*: Shunlin Liang (shunlin@hku.hk)

**Abstract.** Motivated by the lack of long-term global soil moisture products with both high spatial and temporal resolutions, a global 1-km daily spatiotemporally continuous soil moisture product (GLASS SM) was generated from 2000 to 2020 using an ensemble learning model (eXtreme Gradient Boosting－XGBoost). The model was developed by integrating multiple datasets, including albedo, land surface temperature, and leaf area index products from the Global Land Surface Satellite (GLASS) product suite, as well as the European reanalysis (ERA5-Land) soil moisture product, in situ soil moisture dataset from the International Soil Moisture Network (ISMN), and auxiliary datasets (Multi-Error-Removed Improved-Terrain DEM and SoilGrids). Given the relatively large scale differences between point-scale in situ measurements and other datasets, the triple collocation (TC) method was adopted to select the representative soil moisture stations and their measurements for creating the training samples. To fully evaluate the model performance, three validation strategies were explored: random, site-independent, and year-independent. Results showed that although the XGBoost model achieved the highest accuracy on the random test samples, it was clearly a result of model overfitting. Meanwhile, training the model with representative stations selected by the TC method could considerably improve its performance for site- or year-independent test samples. The overall validation accuracy of the model trained using representative stations on the site-independent test samples, which was least likely to be overfitted, was a correlation coefficient (R) of 0.715 and root mean square error (RMSE) of 0.079 $m^3$ $m^{-3}$. Moreover, compared to the model developed without station filtering, the validation accuracies of the model trained with representative stations improved

significantly on most station, with the median R and unbiased RMSE (ubRMSE) of the model for each station increasing from 0.64 to 0.74, and decreasing from 0.055 to 0.052 m$^3$ m$^{-3}$, respectively. Further validation of the GLASS SM product across four independent soil moisture networks revealed its ability to capture the temporal dynamics of measured soil moisture (R = 0.69–0.89; ubRMSE = 0.033–0.048 m$^3$ m$^{-3}$). Lastly, the inter-comparison between the GLASS SM product and two global microwave soil moisture datasets—the 1-km Soil Moisture Active Passive/Sentinel-1 L2 Radiometer/Radar soil moisture product and the European Space Agency Climate Change Initiative combined soil moisture product at 0.25°—indicated that the derived product maintained a more complete spatial coverage, and exhibited high spatiotemporal consistency with those two soil moisture products. The annual average GLASS SM dataset from 2000 to 2020 can be freely downloaded from https://doi.org/10.5281/zenodo.7172664 (Zhang et al., 2022a), and the complete product at daily scale is available at http://glass.umd.edu/soil_moisture/.

## 1 Introduction

Soil moisture typically refers to the water content of the unsaturated soil zone (Liang and Wang, 2020). As an essential climate variable specified by the Global Climate Observing System, it plays a critical role in terrestrial water, energy, and carbon cycles (Dorigo et al., 2017; Humphrey et al., 2021). Over recent decades, soil moisture datasets have been used across a wide range of earth system applications, including climate-related research (Berg and Sheffield, 2018), hydrological modeling (Brocca et al., 2017), rainfall estimating (Brocca et al., 2019), disaster forecasting (Kim et al., 2019), as well as agriculture and ecosystem monitoring (Liu et al., 2020; Holzman et al., 2014), mainly attributed to the progress in remotely sensed soil moisture algorithms. However, substantial gaps remain between the currently released soil moisture products and the growing requirements of various applications, especially at regional and local scales (Peng et al., 2021).

Global soil moisture products can generally be obtained through model simulations or remote sensing, mostly at spatial resolutions of tens of kilometers. The advantages of simulated or reanalysis soil moisture datasets, such as the land component of the European ReAnalysis V5 (ERA5-Land) and the Global Land Data Assimilation System (GLDAS) soil moisture products (Rodell et al., 2004; Muñoz-Sabater et al., 2021), are their spatiotemporal continuity and availability of root-zone estimates; however, their corresponding errors can be rather large when the quality of forcing datasets or model performance are relatively poor (Sheffield et al., 2004). Alternatively, microwave remote sensing has been regarded as the most promising technique to acquire surface soil moisture estimates at global scale, because of its high sensitivity to soil water content dynamics and its capacity for all-weather monitoring (Babaeian et al., 2019; Shi et al., 2019).

Currently, several global soil moisture products have been operationally generated from microwave scatterometers and radiometers, including the Advanced Scatterometer (ASCAT), Advanced Microwave Scanning Radiometer for Earth Observing System (AMSR-E), in addition to instruments on-board the Soil Moisture and Ocean Salinity (SMOS) and Soil Moisture Active Passive (SMAP) satellites (Chan et al., 2016; Wagner et al., 2013; Njoku et al., 2003; Kerr et al., 2016), typically with a grid spacing of 9–50 km, and a

revisit cycle of 1–3 days.

Motivated by the lack of high spatial resolution soil moisture products capable of benefiting numerous regional-scale applications (Peng et al., 2021), various algorithms have been proposed to downscale the more coarse global soil moisture products mentioned above (Peng et al., 2017), some of which have been used to derive global or regional soil moisture products at fine scales in recent years. For example, by combing

Sentinel-1 synthetic aperture radar (SAR) dataset with the SMAP radiometer dataset, Das et al. (2019) generated global soil moisture products at 3 km and 1 km resolutions. Song et al. (2022) downscaled the AMSR-E/AMSR-2 soil moisture products using optical reflectance from the Moderate Resolution Imaging Spectroradiometer (MODIS) and gap-filled land surface temperature (LST) datasets, producing a 1-km daily soil moisture product over China under all-weather conditions. Elsewhere, Naz et al. (2020) generated a daily

soil moisture reanalysis dataset (ESSMRA) at 3 km resolution over Europe by assimilating the European Space Agency (ESA) Climate Change Initiative (CCI) product into a community land model via an ensemble Kalman filter method. Additionally, Vergopolan et al. (2021) recently released a 30 m sub-daily soil moisture dataset across the conterminous United States (CONUS), which was retrieved using the merged 30-m brightness temperatures obtained by combining a hyper-resolution land surface model (HydroBlocks), a

radiative transfer model, and the SMAP Enhanced Level 3 brightness temperatures at 9 km. Apart from these downscaled high-resolution datasets, several studies have directly derived the 1-km operational soil moisture products over Europe from multi-temporal Sentinel-1 SAR images using change detection algorithms, showing potential for global coverage (Balenzano et al., 2021; Bauer-Marschallinger et al., 2019).

Table 1 lists the spatial and temporal coverages, temporal resolution and grid spacing (i.e., pixel size, which

may be finer than the actual spatial resolution) of several representative and publicly available soil moisture products. Accordingly, there remains a lack of long-term global soil moisture products at both high spatial and temporal resolutions. Although the SMAP/Sentinel-1 L2 Radiometer/Radar soil moisture dataset (SPL2SMAP_S) has global coverage and a spatial resolution up to 1 km, its temporal resolution degrades to 12 days over most regions owing to the relatively long revisit cycle of Sentinel-1 SAR satellites. Recently,

Zheng et al. (2023) developed a global seamless soil moisture dataset by downscaling the 0.25° ESA CCI product using random forest model, achieving an R of 0.89 and ubRMSE of 0.045 $m^3$ $m^{-3}$, but they only adopted a random cross-validation strategy which is likely to be affected by model overfitting. Other downscaled soil moisture datasets generally maintain regional or continental coverage, limited by the lack of high-resolution seamless input datasets or model applicability. Optical and thermal remote sensing techniques

can provide long-term observations with high spatiotemporal resolutions, which have been widely used to derive soil moisture or relevant indices (Yue et al., 2019; Ghulam et al., 2007; Rahimzadeh-Bajgiran et al., 2013). However, optical and thermal satellite datasets can be detrimentally affected by cloud coverage, hindering their use in soil moisture retrieval or downscaling across a global scale. To address this issue, the latest versions of Global Land Surface Satellite (GLASS) products (Liang et al., 2021) were used here,

including the spatiotemporally continuous surface albedo, leaf area index (LAI), and land surface temperature (LST), which were produced with reliable accuracies primarily based on MODIS observations. In the present study, these fine-scale GLASS products were integrated with auxiliary datasets (terrain and soil texture) and the seamless ERA5-Land reanalysis soil moisture product at a coarse scale using an ensemble machine-learning model to estimate daily soil moisture at 1 km resolution. This framework was adapted from Zhang

et al. (2022b), where models were trained using Landsat 8 observations and multi-source datasets as inputs, and the International Soil Moisture Network (ISMN) measurements as the target. To produce a seamless global soil moisture product, Landsat datasets prone to cloud interference were replaced with spatiotemporally continuous GLASS products. Considering the larger scale difference between GLASS products and in situ soil moisture compared to Landsat datasets, the triple collocation (TC) technique

(Stoffelen, 1998; McColl et al., 2014) was adopted to select the representative soil moisture stations prior to model training for mitigating the influence of scale mismatch on prediction accuracy.

     Specifically, the aim of this research was to provide a long-term (2000–2020) global soil moisture dataset (GLASS SM) with high spatiotemporal resolutions (1 km, daily) and reliable accuracy. To achieve this goal, an ensemble learning model, eXtreme Gradient Boosting (XGBoost) (Friedman, 2001; Chen and Guestrin,

2016), was developed by integrating multi-source datasets. The model was then applied to generate the global 1-km GLASS SM product, which was further evaluated against four independent soil moisture networks. Lastly, an inter-comparison was made between the derived product and two global microwave soil moisture products to investigate their spatiotemporal consistency.

**Table 1** Main characteristics of several representative and publicly available soil moisture products.

| Category | Soil moisture products | Grid spacing | Spatial coverage | Temporal resolution | Temporal coverage | References | Data link | Notes |
|---|---|---|---|---|---|---|---|---|
| Downscaled products | SPL2SMAP_S | 1/3 km | Global | 6–12 days | 2015–present | Das et al. (2019) | https://nsidc.org/data/spl2smap_s | - |
| | Downscaled ESA-CCI SSM | 1 km | Global | Daily | 2000–2020 | Zheng et al. (2023) | https://doi.org/10.11888/RemoteSen.tpdc.272760 | Seamless |
| | Downscaled AMSR SM | 1 km | China | Daily | 2003–2019 | Song et al. (2022) | http://dx.doi.org/10.11888/Hydro.tpdc.271762 | - |
| | Downscaled ASCAT SM | 1 km | Europe | 1.5 days | 2007–present | Wagner et al. (2008) | https://hsaf.meteoam.it/ | - |
| | ESSMRA | 3 km | Europe | Daily | 2000–2015 | Naz et al. (2020) | https://doi.org/10.1594/PANGAEA.907036 | Seamless |
| | SMAP-HydroBlocks | 30 m | CONUS | 6 hours | 2015–2019 | Vergopolan et al. (2021) | https://doi.org/10.5281/zenodo.5206725 | - |
| Microwave remote sensing products | Sentinel-1 | 1 km | Southern Italy | 6–12 days | 2015–2018 | Balenzano et al. (2021) | https://doi.org/10.5281/zenodo.5006307 | - |
| | CGLS Sentinel-1 SSM | 1 km | Europe | 1.5–8 days | 2014–present | Bauer-Marschallinger et al. (2019) | https://land.copernicus.eu/global/products/ssm | - |
| | ASCAT | 12.5/25 km | Global | Daily | 2007–present | Bartalis et al. (2007) | https://hsaf.meteoam.it/ | - |
| | AMSR-E /AMSR2 | 25 km 10/25 km | Global | Daily | 2002–2011 2012–present | Owe et al. (2008) | https://search.earthdata.nasa.gov/search | - |
| | Fengyun-3 | 25km | Global | Daily | 2011–2020 | Yang et al. (2012) | http://satellite.nsmc.org.cn/ | - |
| | SMAP-L3 | 36 km | Global | Daily | 2015–present | O'Neill et al. (2021) | https://nsidc.org/data/SPL3SMP/versions/8 | - |
| | SMAP-IB | 36 km | Global | Daily | 2015–2021 | Li et al. (2022) | https://ib.remote-sensing.inrae.fr/ | - |
| | SMOS CATDS Level 3 | 25 km | Global | Daily | 2010–present | Al Bitar et al. (2017) | https://www.catds.fr/sipad/ | - |
| | SMOS-IC | 25 km | Global | Daily | 2010–2021 | Wigneron et al. (2021) | https://ib.remote-sensing.inrae.fr/ | - |
| | SGD-SM | 0.25° | Global | Daily | 2013–2019 | Zhang et al. (2021) | https://doi.org/10.5281/zenodo.4417458 | Seamless |
| | MCCA-AMSR MCCA-SMAP | 0.25° 36 km | Global | Daily | 2002–2021 2015–2022 | Zhao et al. (2021) | https://doi.org/10.11888/Terre.tpdc.272907 https://doi.org/10.11888/Terre.tpdc.272088 | - |
| | ESA CCI | 0.25° | Global | Daily | 1978–2021 | Gruber et al. (2019) | https://esa-soilmoisture-cci.org/data | - |
| Reanalysis products | GLDAS-Noah | 0.25° | Global | 3 hours | 2000–2021 | Beaudoing and Rodell (2020) | https://hydro1.gesdisc.eosdis.nasa.gov/data/GLDAS/GLDAS_NOAH025_3H.2.1/ | Seamless |
| | ERA5-Land | 0.1° | Global | Hourly | 1950–present | Muñoz-Sabater (2019, 2021) | https://cds.climate.copernicus.eu/cdsapp#!/dataset/reanalysis-era5-land | Seamless |
| Present study | GLASS SM | 1 km | Global | Daily | 2000–2020 | - | http://glass.umd.edu/soil_moisture/ | Seamless |

## 2 Datasets

The multi-source datasets used to generate the global high-resolution soil moisture product here can be grouped into four categories (Table 2). Namely, remotely sensed variables derived from the three GLASS products, reanalysis surface soil moisture from ERA5-Land dataset, and auxiliary variables extracted from

the Multi-Error-Removed Improved-Terrain (MERIT) DEM and SoilGrids products were used to train an XGBoost model for estimating the global soil moisture product; whereas globally distributed in situ soil

moisture measurements from ISMN stations were used as targets for model training. In addition, four independent in situ soil moisture datasets, and two microwave soil moisture products were used to validate and compare the derived global product.

**Table 2** Multi-source datasets used to generate the global high-resolution soil moisture product.

| Category | Dataset | Spatial resolution | Temporal resolution |
|---|---|---|---|
| Satellite products | GLASS albedo | 500 m | 4-day |
| | GLASS LST | 1 km | Daily |
| | GLASS LAI | 500 m | 8-day |
| Reanalysis product | ERA5-Land SSM | 0.1° | Hourly |
| Auxiliary datasets | MERIT DEM | 90 m | - |
| | SoilGrids 2.0 | 250 m | - |
| Ground-based data | ISMN SSM | Point scale | Hourly |

**2.1 Remotely sensed datasets**

The GLASS product suite has been employed in various applications owing to its long-term coverage, spatial continuity, high spatial resolution, and accuracy (Liang et al., 2021). Here, the latest version of GLASS albedo, LST, and LAI products served as the primary inputs to the ensemble learning model. Specifically, the GLASS V6 LAI product (500 m resolution) was generated from six MODIS 8-day surface reflectance bands of MOD09A1 using a bidirectional long short-term memory deep learning model ([www.glass.umd.edu](www.glass.umd.edu)) (Ma and Liang, 2022). Notably, this product is relatively more accurate than the 250 m GLASS LAI estimated from two bands of MOD09Q1. The all-sky 1-km GLASS LST was produced by integrating multiple datasets from MODIS, reanalysis, and in situ LST measurements using a random forest model (Li et al., 2021). Daily global LSTs averaged from instantaneous GLASS LST products were used here, which can be downloaded soon from [www.glass.umd.edu](www.glass.umd.edu). The gap-free GLASS albedo products were generated using a combination of a direct-estimation algorithm (Qu et al., 2014), and a spatiotemporal filtering scheme (Liu et al., 2013). Namely, the black-sky visible, near-infrared, and shortwave albedo extracted from the GLASS V42 albedo products were used in the present study ([www.glass.umd.edu](www.glass.umd.edu)).

**2.2 ERA5-Land reanalysis soil moisture product**

ERA5 provides a range of global atmospheric, terrestrial, and oceanic variables from 1950 to present at 31 km spatial resolution (Hersbach et al., 2020). Specifically, ERA5-Land is an enhanced global land reanalysis dataset obtained by downscaling the atmospheric forcing derived from the reanalysis of ERA5 to a native resolution of approximately 9 km (Muñoz-Sabater et al., 2021). ERA5-Land includes hourly estimates of volumetric soil moisture at four soil layers, and a grid spacing of 0.1° ([https://cds.climate.copernicus.eu/](https://cds.climate.copernicus.eu/)). In

the present study, the top layer (0–7 cm) of ERA5-Land soil moisture were used to match the shallow observation depths of optical satellites. The daily average soil moisture was calculated and resampled to 1 km before being used as an input variable of the ensemble learning model.

**2.3 Static terrain and soil texture datasets**

Topography and soil properties, which can be treated as static variables due to their relatively slow rate of change over the short term, have an important influence on the spatial variations of soil moisture at finer scales. The global terrain dataset used in the study here was the high-accuracy MERIT DEM with a spatial resolution of 3 arc seconds (~90 m at the equator). The MERIT DEM integrates several spaceborne DEMs after eliminating their inherent primary error components, including speckle noise, stripe noise, absolute bias,

and tree height bias (http://hydro.iis.u-tokyo.ac.jp/~yamadai/MERIT_DEM/) (Yamazaki et al., 2017). After deriving the elevation, aspect, and slope from the MERIT DEM, these topographic variables were resampled to 1 km, and used as input features for the model. Alternatively, soil texture was derived from the SoilGrids V2.0 product at 250 m resolution (https://www.isric.org/explore/soilgrids). SoilGrids uses > 240,000 soil profile measurements, and > 400 environmental covariates worldwide to train machine learning models, and

produce global soil property maps across six depth intervals (Poggio et al., 2021). Recent studies have shown that the SoilGrids product has both higher resolution and enhanced accuracy compared to other soil datasets at global scale (Dai et al., 2019), in addition to the ability of soil texture data to improve the bias and root mean square error (RMSE) of downscaled soil moisture products (Das et al., 2019). Accordingly, the mean contents of sand, silt, and clay were extracted for the first soil layer (0–5 cm) from the SoilGrids database,

and resampled to 1 km.

**2.4 Ground-based soil moisture training dataset**

The ISMN aims to establish and maintain a global database of in situ soil moisture measurements for the validation and improvement of satellite-based and modelled soil moisture products. Currently, it consists of 73 networks with over 2800 soil moisture stations worldwide, providing quality-controlled and harmonized

datasets collected from monitoring networks and field experiments (Dorigo et al., 2021). Here, data for the period from 2000–2018 were obtained (https://ismn.earth/), and only stations with a sensing depth of < 5 cm were selected to match the observation depth of remotely sensed datasets. Soil moisture records were then screened according to the quality flags provided with the ISMN dataset (Dorigo et al., 2013), before being used as the training target for the machine learning model. The spatial distribution of the representative ISMN

soil moisture stations selected using the TC method described in Sect. 3.2 is displayed in Fig. 1. The number

and percentage of representative stations for each land cover type and climate class, which are calculated by using the 500-m MODIS land cover type product (Friedl and Sulla-Menashe, 2019) and the 1-km Köppen–Geiger climate classification dataset (Cui et al., 2021), respectively, are also shown in Table 3.

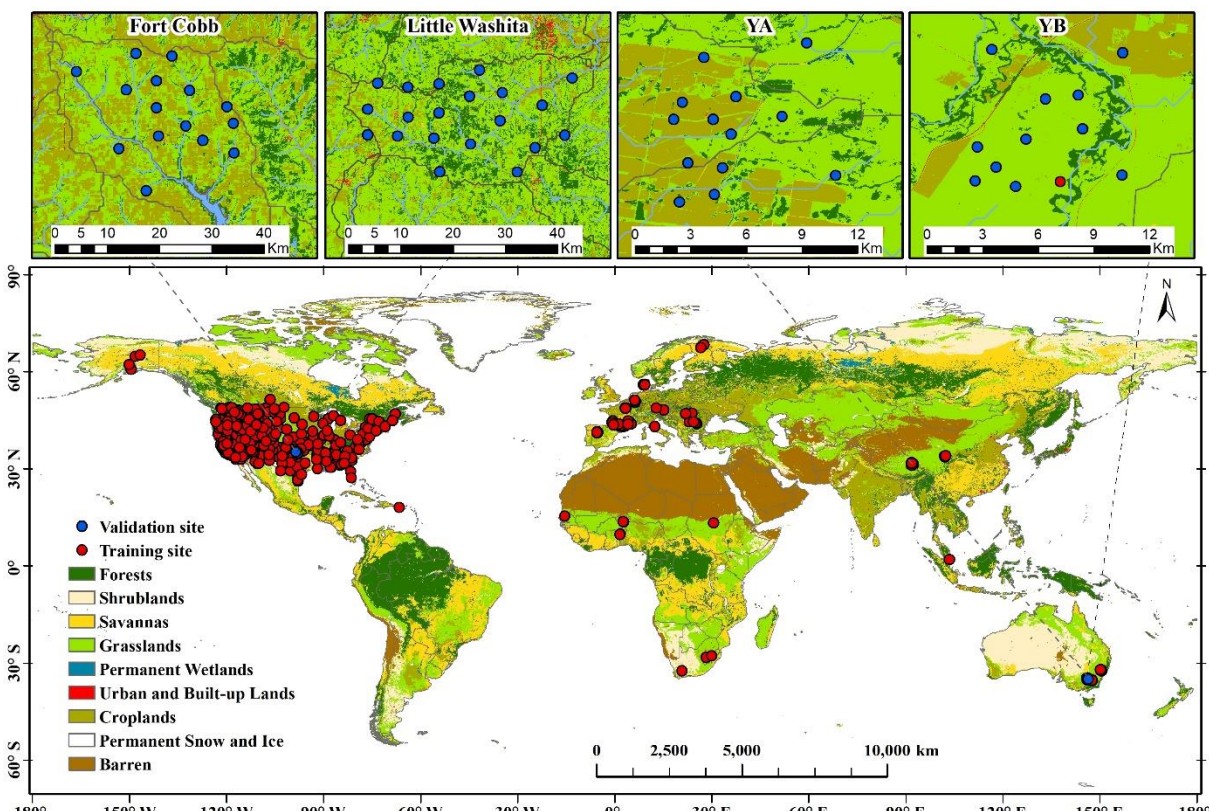

**Figure 1** Spatial distribution of the 715 representative ISMN soil moisture stations used for training the model and four independent soil moisture networks used for validation, with the MODIS land cover type product (MCD12Q1) for 2016 displayed in the background.

**Table 3** The number and percentage of representative ISMN soil moisture stations for each climate class and land cover type.

| Climate class | Num | % | Land cover type | Num | % |
|---|---|---|---|---|---|
| Tropical | 8 | 1.1 | Forests | 35 | 4.9 |
| Arid | 135 | 18.9 | Shrublands | 16 | 2.2 |
| Temperate, dry summer | 125 | 17.5 | Savannas | 185 | 25.9 |
| Temperate, dry winter | 2 | 0.3 | Grasslands | 327 | 45.7 |
| Temperate, no dry season | 194 | 27.1 | Urban | 12 | 1.7 |
| Cold, dry summer | 36 | 5.0 | Croplands | 130 | 18.2 |
| Cold, dry winter | 6 | 0.8 | Barren | 10 | 1.4 |
| Cold, no dry season | 176 | 24.6 | | | |
| Polar | 33 | 4.6 | | | |

## 2.5 Independent in situ validation datasets

Four soil moisture monitoring networks that were not included in the ISMN database were used to assess

the model's ability to capture temporal variations in soil moisture over unknown area (Fig. 1). The YA and YB subnetworks are both part of the Yanco soil moisture network, located in a semi-arid agricultural region of the Murrumbidgee River Basin, Australia, with a flat topography, and elevation spanning 117–150 m (Yee et al., 2017). There are 13 and 11 stations in the YA and YB subnetworks, respectively, distributed across two 9 × 9 km areas, and soil moisture observations from these stations can be downloaded from the Oznet Hydrological Monitoring website (http://www.oznet.org.au) (Smith et al., 2012). Two other micronets (Fort Cobb and Little Washita) are located in southwestern Oklahoma, USA, and are characterized by a humid subtropical climate (Starks et al., 2014). The primary land cover types are cropland and rangeland, and the topography is moderately rolling (Bindlish et al., 2009). Currently, there are 15 and 20 operational stations in the Fort Cobb and Little Washita networks, respectively, for which soil moisture datasets can be accessed through the Grazinglands Research Laboratory (https://ars.mesonet.org/). These four dense soil moisture networks have been used extensively to either validate or calibrate satellite soil moisture products (Ma et al., 2021; Colliander et al., 2017; Chan et al., 2018).

**2.6 Microwave soil moisture product**

To further validate the spatiotemporal performance of the derived 1-km soil moisture product here, two additional microwave-based products were selected for comparison. The first product is the high resolution SMAP/Sentinel-1 SPL2SMAP_S dataset, which contains the first global 1-km soil moisture product that was publicly released in the past (Table 1). It has a temporal resolution of 6–12 days and can be downloaded from the National Snow and Ice Data Center at 1 km and 3 km resolutions (https://nsidc.org/data/spl2smap_s). According to Das et al. (2019), the average unbiased RMSE (ubRMSE) values achieved by both the 1-km and 3-km SPL2SMAP_S products over sparse soil moisture networks were approximately 0.05 $m^3$ $m^{-3}$. Considering that the SPL2SMAP_S baseline algorithm generally shows higher validation accuracy than the optional algorithm (directly disaggregating the SMAP 9-km soil moisture product), and the AM (descending orbits combination) soil moisture retrievals are more accurate than their APM equivalents (descending or ascending orbits combination) (Xu, 2020), the baseline AM soil moisture field "disagg_soil_moisture_1km" were extracted from the SPL2SMAP_S 1-km data group, and used for comparison. The second product is the CCI global soil moisture dataset released by the ESA, with a grid spacing of 0.25° and daily temporal resolution, which combines various passive and active microwave soil moisture products into a harmonized record with improved spatiotemporal coverages and has been fully validated across numerous global applications (Gruber et al., 2019; Dorigo et al., 2017). Specifically, the combined (active and passive) soil

moisture product from CCI V6.1 was used here (https://esa-soilmoisture-cci.org/data).

## 3 Methods

### 3.1 Overall framework

Soil moisture is characterized by high spatiotemporal variability and its distribution is influenced by a range of environmental factors across different scales, such as climate, geographical conditions, soil properties, and surface coverage (Crow et al., 2012; Luo et al., 2022). Here, high-accuracy, spatiotemporally continuous GLASS products, including LST, albedo, and LAI, were used to provide surface temperature, spectral information on soil and vegetation, as well as information related to vegetation type and density.

Considering the impact of topography and soil properties on soil moisture, topographic and soil texture fraction variables were extracted from the MERIT DEM and SoilGrids products, respectively. Additionally, the 0.1° ERA5-Land reanalysis soil moisture product was used to provide background soil moisture information. By utilizing an ensemble machine learning model, various variables extracted from these multi-source datasets were integrated so that different environmental factors affecting soil moisture could be

accounted for, and then soil moisture at fine scales could be estimated.

Figure 2 shows a flowchart of the proposed 1-km, spatiotemporally continuous soil moisture estimation framework. Prior to the training phase, the TC method and the other two long-term soil moisture datasets (ERA5-Land reanalysis and ESA CCI soil moisture products) were adopted for selecting the representative soil moisture stations, considering the scale difference between point-scale soil moisture measurements

collected by ISMN stations and GLASS products (the detailed selection procedure is presented in Sect. 3.2). Then, multiple variables were extracted from the corresponding input datasets, and spatiotemporally collocated with the in situ soil moisture measurements from the representative stations between 2000 and 2018. Specifically. the black-sky visible, near-infrared, shortwave albedo, LAI, and LST were extracted from the three GLASS products, based on the geographic locations of stations. Each of these variables, together

with topographic and soil texture fraction variables, and the coarse-scale reanalysis soil moisture were put into the XGBoost model, which was chosen to simulate the non-linear relationship between multiple input features and in situ soil moisture (the target variable). Lastly, those multi-source input datasets were resampled to 1 km, and then put into the developed XGBoost model for predicting the global 1-km spatiotemporally continuous soil moisture product (GLASS SM). Moreover, the GLASS SM product was

evaluated against four independent soil moisture datasets, and then compared the SPL2SMAP_S and CCI soil moisture products for spatiotemporal consistency analyses.

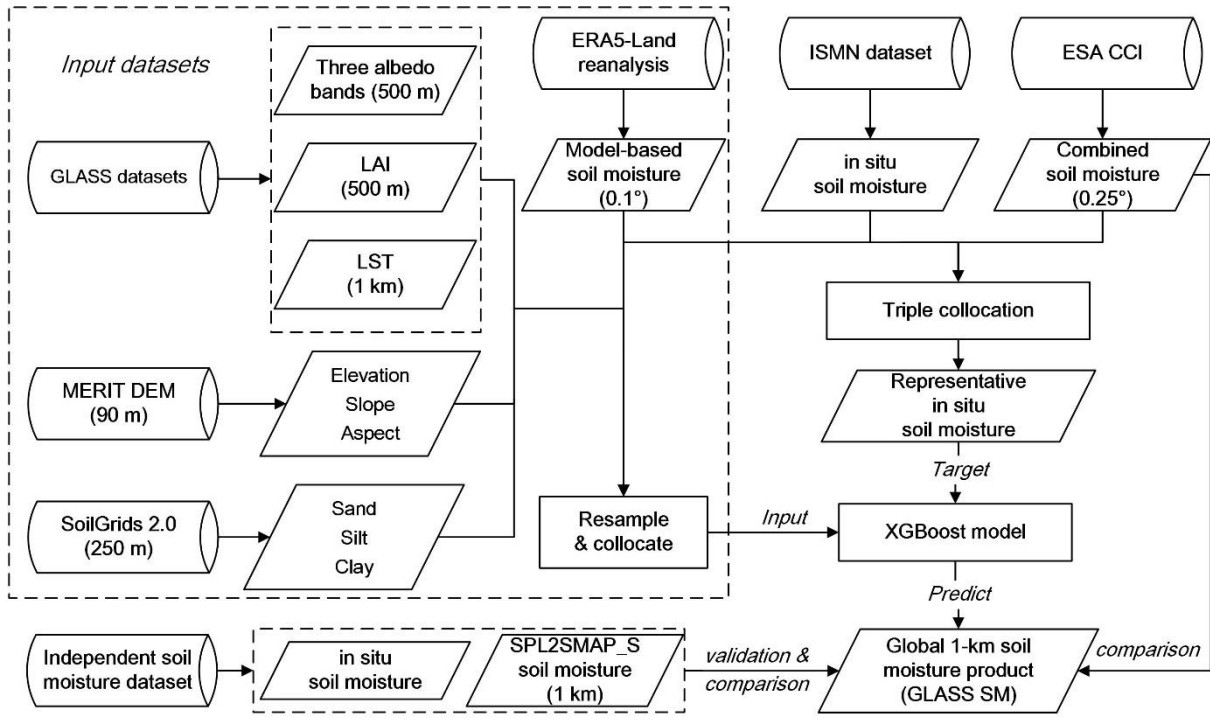

**Figure 2** Flowchart of the proposed 1-km spatiotemporally continuous soil moisture estimation framework.

**3.2 Triple collocation-based station selection**

As mentioned above, in situ soil moisture data from the ISMN stations were employed as the target variable to train the XGBoost model, which was then used to predict soil moisture product at 1 km resolution. The underlying assumption was that the measured soil moisture at these point-scale stations is representative of the average moisture status of the corresponding 1-km pixel; however, because of the high spatiotemporal variability of soil moisture, this assumption is not always upheld. Accordingly, the TC method, which has

been widely applied to analysis the coarse-scale spatial representativeness of in situ soil moisture dataset (Gruber et al., 2013; Molero et al., 2018), was adopted here to select the most representative stations. Specifically, TC is an error analysis method proposed by Stoffelen (1998) employing three collocated datasets to address large uncertainties in wind speed measurements. TC has also been widely used in the evaluation of satellite soil moisture products given the limited number of core validation sites at the satellite footprint

scale (Zheng et al., 2022). The commonly used error model for TC analysis is defined in Eq. (1):

$$X_i = \alpha_i + \beta_i \theta + \varepsilon_i \qquad (1)$$

where $X_i$ refers to the three collocated soil moisture observations; $\theta$ refers to the unknown true value of soil moisture; $\alpha_i$ and $\beta_i$ are the additive and multiplicative biases of $X_i$ relative to the true value, respectively; and $\varepsilon_i$ is the random additive noise with zero mean. The assumptions underlying this error model and detailed derivation process for the error variance of each dataset can be found in Gruber et al.

(2016). Notably, the assumptions made for TC analysis are similar to those made for the correlation

coefficient (R) and RMSE (Gruber et al., 2016). To fulfill the independent error requirement of the TC

analysis across the three datasets, the ISMN in situ soil moisture, model-based ERA5-Land soil moisture,

and CCI combined microwave soil moisture were selected to construct the triplets. Among them, the CCI

soil moisture product was selected here rather than other microwave soil moisture products, as it maintains a

sufficiently long timescale to cover that of the training samples. The error variance of the ISMN soil moisture

dataset, $\sigma_\varepsilon^2$ , was then calculated according to Eq. (2):

$$\sigma_\varepsilon^2 = \sigma_{ismn}^2 - \frac{Cov(X_{ismn}, X_{era})Cov(X_{ismn}, X_{cci})}{Cov(X_{era}, X_{cci})} \tag{2}$$

where $\sigma_{ismn}^2$ is the variance of the ISMN in situ soil moisture; $Cov$ is the covariance operator; and

$X_{ismn}$, $X_{era}$, and $X_{cci}$ denote the collocated ISMN, ERA5-Land, and CCI soil moisture observations,

respectively. Based on TC analysis, McColl et al. (2014) proposed a method called extended triple collocation

(ETC) to estimate the correlation coefficient between each dataset and the unknown target variable.

Specifically, the ETC correlation coefficient of the ISMN soil moisture dataset, $R_{ETC}$, can be calculated via

Eq. (3):

$$R_{ETC} = sign(\pm)\sqrt{\frac{Cov(X_{ismn}, X_{era})Cov(X_{ismn}, X_{cci})}{\sigma_{ismn}^2 Cov(X_{era}, X_{cci})}} \tag{3}$$

where the sign of $R_{ETC}$ was corrected to positive. It is a scaled, unbiased signal-to-noise ratio metric

complementary to $\sigma_\varepsilon^2$. Using the above TC-based metrics, and referring to previous studies (Yuan et al.,

2020; Anderson et al., 2012), several strict conditions were established to select the most representative ISMN

stations: (1) > 500 triplets were available at the station during the period 2000–2018, (2) the R between any

two soil moisture datasets in the triplets was > 0.2, (3) the square root of the $\sigma_\varepsilon^2$ calculated for the ISMN

soil moisture dataset was < 0.06 m$^3$ m$^{-3}$, and (4) the $R_{ETC}$ between the ISMN soil moisture and the unknown

soil moisture true values was > 0.7. A total of 715 representative ISMN soil moisture stations were finally

selected, as shown in Fig. 1.

**3.3 XGBoost model**

Ensemble machine learning models can be roughly classified into two categories based on how the

individual learners are generated: bagging and boosting, (Zhou, 2021). For bagging models, the individual

learners are constructed independently; whereas for boosting models, learners are constructed iteratively,

increasing the weights for the incorrectly classified samples during each round of training. As a representative

bagging algorithm, random forest has gained considerable attention in the fields of remote sensing classification and regression over recent decades (Belgiu and Drăguţ, 2016); however, it may suffer from a large prediction bias, especially when the observations are too large or small (Song, 2015). In contrast, boosting models have been shown to reduce both variance and bias and are robust to multicollinearity among predictors (Gislason et al., 2006; Karthikeyan and Mishra, 2021). Accordingly, the present study employed the XGBoost model implemented by Chen and Guestrin (2016) based on a gradient boosting framework (Friedman, 2001). The XGBoost model is advantageous for its scalability, efficiency, and decreased vulnerability to overfitting. Here, the open-source *xgboost* and *Scikit-learn* Python packages were used together for model training and hyperparameters tuning, with the grid search method being adopted to determine the optimal parameters. Here, the key hyperparameters of the XGBoost models were finally set to n_estimators (the number of the boosting rounds) = 1000, learning_rate = 0.1, and max_depth (maximum tree depth) = 8.

**3.4 Evaluation strategies and performance metrics**

While most previous soil moisture estimation studies based on machine learning have only used the random validation approach, this study used the three complementary validation strategies to fully evaluate the model performance: random, site-independent, and year-independent. For the random validation, samples from all soil moisture stations during 2000–2018 were randomly divided into five folds, among which three folds were used for training, one as the validation dataset to tune the hyperparameters of the model, and one as the test dataset to evaluate the model performance. Thus, the samples in the random test dataset may have been from the same station or year as the training or validation datasets. For site-independent validation, all soil moisture stations were again randomly divided into five folds, and samples from one fold were used as the test dataset to evaluate the accuracy of models trained with samples from the other folds, which were used for training and validation. Thus, the location of the samples in the site-independent test dataset is unknown to the model. Similarly, for the year-independent validation, samples from all stations between 2015 and 2018 were selected as the test dataset to evaluate the accuracy of the model trained using samples between 2000 and 2014, to ensure that the observation year was unknown to the model.

In addition to model evaluation, the accuracy of the GLASS SM product generated by the developed model was evaluated. This 1-km soil moisture product was first validated against four independent dense soil moisture networks, and then compared with the 1-km SPL2SMAP_S and 0.25° CCI soil moisture products for spatiotemporal consistency analyses. Four widely used performance metrics in soil moisture related

researches—the R, bias, RMSE, and ubRMSE (Entekhabi et al., 2010) are used to evaluate both the models

and products against in situ dataset, which can be calculated according to Eqs.( 4–7):

$$R = \frac{E[(\theta_{est} - E[\theta_{est}])(\theta_{true} - E[\theta_{true}])]}{\sigma_{est}\sigma_{true}} \tag{4}$$

$$bias = E[\theta_{est}] - E[\theta_{true}] \tag{5}$$

$$RMSE = \sqrt{E[(\theta_{est} - \theta_{true})^2]} \tag{6}$$

$$ubRMSE = \sqrt{E\{[(\theta_{est} - E[\theta_{est}]) - (\theta_{true} - E[\theta_{true}])]^2\}} \tag{7}$$

where $E[.]$ denotes the mean operator; $\theta_{true}$ and $\theta_{est}$ represent the in situ soil moisture and

corresponding estimated soil moisture; whereas $\sigma_{true}$ and $\sigma_{est}$ refer to the standard deviation of the in

situ and estimated soil moisture values, respectively. Note that, while comparing two soil moisture products

with similar spatial resolution in Sect. 4,4, the term "root mean square difference (RMSD)" is used, despite

that it is also calculated using Eq. (6). Besides, when large-scale soil moisture product is validated against

point-scale in situ soil moisture dataset, bias often exist between the two datasets because of scale differences,

and then R and ubRMSE are typically more informative than RMSE.

## 4 Results

In Sect. 4.1, the overall performance of the XGBoost models trained using different groups of stations was

first evaluated using random test samples. Then, the performance of the models was evaluated on the site- or

year-independent test samples in Sect. 4.2, where the permutation feature importance results of the models

and the importance of each type of input variables were examined, followed by an analysis of the model

performance metrics at each station and over each land cover type. Sect. 4.3 shows the time-series validation

results of the GLASS SM product generated using the developed model on four independent soil moisture

networks; whereas Sect. 4.4 compares the global 1-km GLASS SM product with two global microwave soil

moisture products for spatiotemporal consistency analyses.

### 4.1 Model performance on the random test samples

Figure 3 shows the overall performance of the XGBoost models developed using all input variables on the

random test samples. To analyze the effect of screening soil moisture stations, the accuracies of models

developed using all ISMN stations, the representative stations selected using the TC method, and the stations

excluded using the TC method (not included in the representative stations) were compared via scatterplots.

In general, the random validation accuracy of all three XGBoost models was high, with the bias between the

model-predicted and target soil moisture values being close to zero. The accuracy of the models developed

using all ISMN stations or the TC-excluded stations were similar for the test samples, with R values of 0.917 and 0.918, and RMSE values of 0.047 $m^3$ $m^{-3}$ and 0.049 $m^3$ $m^{-3}$, respectively. In contrast, the accuracy of the model developed with the representative stations selected using the TC method was significantly improved for the test samples, with R and RMSE values of 0.941 and 0.038 $m^3$ $m^{-3}$, respectively. Compared with the other two models, the soil moisture estimates of the XGBoost model developed using representative stations were more concentrated along the 1:1 line. Notably, most of the soil moisture measurements that were nearly saturated (> 0.5 $m^3$ $m^{-3}$) were excluded after the station screening process (Fig. 3), likely because those high soil moisture samples at point-scales were typically under-representative of the mean soil moisture conditions at satellite footprint-scales. Meanwhile, the validation accuracy of the ERA5-Land surface soil moisture product was also calculated for all soil moisture samples, as well as those selected by the TC method for comparison. After station screening, the overall R between ERA5-Land reanalysis and in situ soil moisture increased from 0.56 to 0.64, while the RMSE decreased slightly from 0.138 to 0.129 $m^3$ $m^{-3}$ and the bias remained unchanged at 0.08 $m^3$ $m^{-3}$. Above performance metrics indicated that representative stations can be effectively selected by using the TC method, and training the XGBoost model with representative stations can significantly improve its validation accuracy on the random test samples.

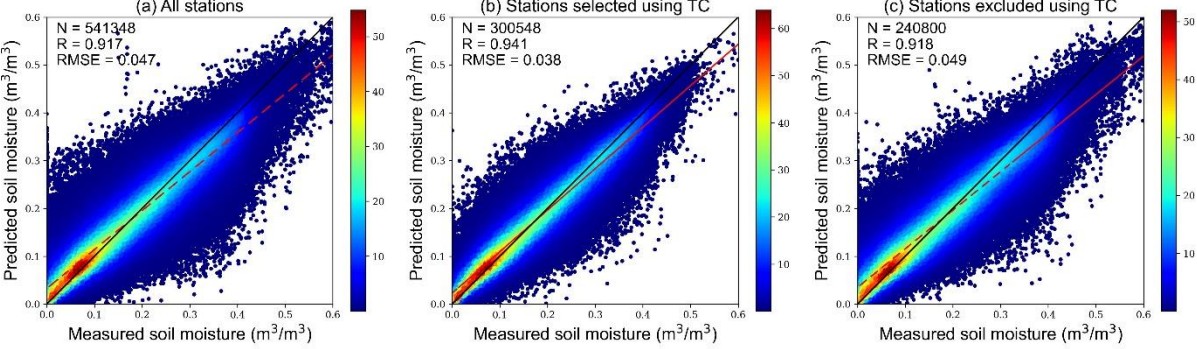

**Figure 3** Scatterplots of measured and predicted soil moisture from the XGBoost models developed using (a) all ISMN stations, (b) representative stations selected using the TC method, and (c) stations excluded using the TC method. Point colors indicate the probability density. Red dotted line displays the linear regression, and the black solid line is the 1:1 line.

### 4.2 Model performance on site/year independent samples

As can be seen from Table 4, regardless of the type of soil moisture station used during training, model performance on the year-independent test samples (2015 to 2018) decreased significantly compared to that on the random test samples. Among them, the R values of the models trained using all stations and TC-excluded stations were 0.8 and 0.734 for the year-independent test samples, respectively, while the

corresponding RMSE increased to 0.07 and 0.084 $m^3$ $m^{-3}$, respectively. In contrast, the XGBoost model trained using representative stations selected by the TC method achieved the highest accuracy on the year-independent test samples, with R and RMSE values of 0.873 and 0.054 $m^3$ $m^{-3}$, respectively. Likewise, the performance of the models trained using three different types of stations on the site-independent test samples (randomly selected one-fifth of the total stations) further decreased compared to that of the year-independent test samples. The RMSE of the models trained using all and excluded stations further increased to 0.093 and 0.106 $m^3$ $m^{-3}$, respectively, for the site-independent test samples. Alternatively, the XGBoost model trained using representative stations achieved the highest accuracy for the site-independent test samples, with R and RMSE values of 0.715 and 0.079 $m^3$ $m^{-3}$, respectively. These results suggest that the good performance of the models on the random or year-independent test samples is clearly a result of model overfitting, and their accuracies may degrade significantly when the stations or observation years of the test samples are unknown to them. While the relatively lower accuracy achieved by the model on site-independent test samples is least likely to be overfitted and can be regarded as the model's true accuracy. Besides, it appears that increasing the number of stations in the training dataset to account for spatial heterogeneity is more important for improving the models' performance than extending the duration of the measurements to account for temporal dynamics, as also found in previous study (Zappa et al., 2019). Moreover, training the model with representative stations selected by the TC method can also considerably improve its performance on site- or year-independent test samples, that is, model performance over unknown time and space.

**Table 4** Validation accuracy of the XGBoost models trained using three different types of soil moisture stations on three types of test samples.

| Validation strategies | All stations | | | Representative stations | | | Excluded stations | | |
|---|---|---|---|---|---|---|---|---|---|
| | R | RMSE ($m^3$ $m^{-3}$) | ubRMSE ($m^3$ $m^{-3}$) | R | RMSE ($m^3$ $m^{-3}$) | ubRMSE ($m^3$ $m^{-3}$) | R | RMSE ($m^3$ $m^{-3}$) | ubRMSE ($m^3$ $m^{-3}$) |
| Random | 0.917 | 0.047 | 0.047 | 0.941 | 0.038 | 0.038 | 0.918 | 0.049 | 0.049 |
| Year-independent | 0.800 | 0.070 | 0.070 | 0.873 | 0.054 | 0.054 | 0.734 | 0.084 | 0.084 |
| Site-independent | 0.630 | 0.093 | 0.093 | 0.715 | 0.079 | 0.079 | 0.564 | 0.106 | 0.106 |

Figure 4 shows the permutation feature importance results of the XGBoost models trained using representative soil moisture stations, which were calculated separately for the three different types of test samples. The permutation importance of an input feature is commonly measured by the degradation of model accuracy when the feature is randomly shuffled (Breiman, 2001), can be calculated multiple times across a test dataset and is less likely to be biased towards high-cardinality features. Notably, permutation importance does not reflect a feature's intrinsic predictive value, but rather its relative importance to a particular model.

For all three types of test samples, ERA5-Land surface soil moisture (SM_era) achieved the highest importance score, indicating that this coarse-scale reanalysis soil moisture product can indeed provide reliable soil moisture background information for the 1-km soil moisture estimation model. Specifically, for both the random and year-independent test samples (Fig. 4 (a), (b)), the importance of elevation and soil texture variables (sand, silt, and clay) ranked relatively high, showing that soil properties and topographic factors are important for accurate model predictions when the sample locations are known. In addition, the three GLASS black-sky albedo bands (ABD_vis, ABD_nir, and ABD_short) also achieved relatively high importance scores for both types of samples, likely because surface albedo can reflect the surface energy flux and land cover conditions, which are further correlated to the spatial variation in soil moisture (Long et al., 2019). Meanwhile, the importance scores of GLASS LAI and LST were relatively low for the two sample types, which may be partly attributed to their correlation with some high-ranking variables (e.g., ABD_vis, SM_era). For example, after removing ERA5-Land soil moisture from the models, the importance scores of both GLASS LST and LAI increased significantly. In contrast, for the site-independent test samples (Fig. 4 (c)), the importance of ERA5-Land surface soil moisture (SM_era) further increased relative to other variables. In addition, the importance ranking of GLASS albedo and LST increased remarkably; whereas that of terrain and soil texture-related variables dropped dramatically, suggesting that when the location of the test samples is unknown to the model, variables such as coarse-scale soil moisture, albedo, and LST appear to be more important for accurately predicting soil moisture. Note that the final model was developed using all the representative ISMN stations, and its feature importance results over unknown regions could refer to those calculated on the site-independent test samples.

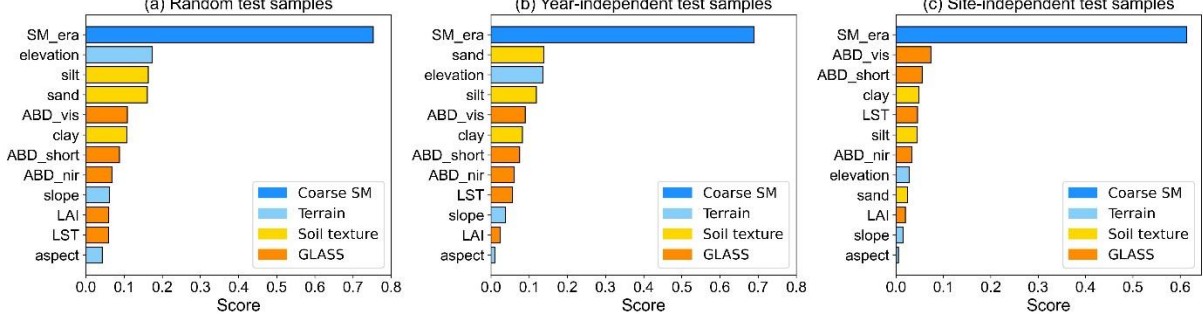

**Figure 4** Permutation feature importance results of the XGBoost models trained using the representative stations, and calculated using the (a) Random, (b) Year-independent, and (c) Site-independent test samples. Features from different input datasets are divided into four groups with different colors.

To further investigate the importance of different types of input variables for the 1-km soil moisture

estimation model over unknown space, the validation accuracy of the XGBoost models developed using different combinations of input datasets on the site-independent test samples were also compared. The XGBoost model trained with all input datasets achieved the highest accuracy (Table 5), with R and RMSE values of 0.715 and 0.079 $m^3$ $m^{-3}$, respectively. After the ERA5-Land soil moisture product was excluded, the model accuracy for the test dataset decreased significantly, with the RMSE value increasing to 0.086 $m^3$ $m^{-3}$, further reflecting the relatively high importance of the coarse-scale soil moisture background information for the 1-km estimation model derived here. Similarly, after excluding GLASS albedo, LAI, and LST from the input variables, the model trained with the remaining variables showed a marked decrease in accuracy for the test dataset, with R and RMSE values of 0.694 and 0.083 $m^3$ $m^{-3}$, respectively. This indicates that the information on soil and vegetation reflective properties, surface temperature, as well as vegetation types and densities provided by GLASS products are also important for the 1-km soil moisture estimation model. Further, the exclusion of terrain or soil texture datasets showed a similar effect on model accuracy, with RMSE values decreasing to 0.082 and 0.083 $m^3$ $m^{-3}$, respectively, again suggesting the pertinent contribution of these variables to improving the performance of the soil moisture estimation model. Besides, as shown in Table 2, the spatial resolution of most input datasets was within 1 km, except for the ERA5-Land product which had a relatively low spatial resolution (0.1°). Therefore, the integration of multi-source input datasets using a machine learning model can improve not only the model accuracy, but the spatial details of the soil moisture product as well. Because the XGBoost model trained with all input datasets performed best on the test dataset, all datasets were included in model training during the subsequent experiments.

**Table 5** Performance metrics of the XGBoost model developed using different combinations of input datasets on the site-independent test samples.

| Input datasets | R | RMSE ($m^3$ $m^{-3}$) | ubRMSE ($m^3$ $m^{-3}$) |
|---|---|---|---|
| All datasets included | 0.715 | 0.079 | 0.079 |
| Coarse SM (ERA5-Land) excluded | 0.646 | 0.086 | 0.086 |
| Albedo, LAI & LST (GLASS) excluded | 0.694 | 0.083 | 0.082 |
| Terrain (MERIT) excluded | 0.700 | 0.082 | 0.082 |
| Soil texture (SoilGrids) excluded | 0.684 | 0.083 | 0.083 |

To explore the causes of decreased 1-km soil moisture estimation model accuracies over unknown time and space, performance metrics of the models were calculated for each station, which were trained using all ISMN or representative soil moisture stations selected by the TC method. To obtain the validation accuracy for each station, a 5-fold cross-validation method was adopted, where the stations were randomly divided

into five folds, with samples from four folds used to develop the model, and the accuracy metrics were derived for the remaining fold. This process was repeated five times, until the accuracies of all stations were evaluated. The distribution of performance metrics for the XGBoost model developed using all stations was dispersed across stations, with R values ranging from -1 to 1, and RMSE values ranging from 0.005 to 0.397 $m^3$ $m^{-3}$ (Fig. 5, Table 6). Although the median of the bias between model predicted and measured soil moisture was 0, the model exhibited a large prediction bias for most stations (from -0.39 to 0.34 $m^3$ $m^{-3}$), partly contributing to the large RMSE observed at these stations. After removing the prediction bias for each station, the median ubRMSE of the model decreased to 0.055 $m^3$ $m^{-3}$, compared to the median RMSE of 0.075 $m^3$ $m^{-3}$. As a comparison, the performance metrics of the ERA5-Land soil moisture product at each ISMN station were also calculated and displayed in Fig. 5. The coarse-scale soil moisture product showed similar R values to those of the XGBoost model developed using all stations, and it also yielded large bias and dispersed RMSE and ubRMSE values at most stations.

After filtering the stations using the TC method, the accuracies of the ERA5-Land soil moisture product at those representative stations improved significantly. Similarly, the validation accuracies of the model developed using the representative stations also improved significantly, with the distribution of its performance metrics being more concentrated across stations, compared to the model developed without station filtering. In particular, the median R of the model at each station increased from 0.64 to 0.74, median RMSE decreased from 0.075 to 0.068 $m^3$ $m^{-3}$, and ubRMSE decreased from 0.055 to 0.052 $m^3$ $m^{-3}$. Over most of the representative stations, the XGBoost model obtained similar or even larger R values compared to the ERA5-Land soil moisture product. However, there were also several stations where the model achieved relatively lower R values, yet this degradation in temporal metrics with respect to the original coarse-scale products can be found in many soil moisture downscaling studies (Gruber et al., 2020).

On the other hand, the model developed using the representative stations still exhibited a large bias at most stations, ranging from -0.21 to 0.21 $m^3$ $m^{-3}$, although the median bias of the model was 0. Therefore, the decreased overall accuracies of the model over unknown spaces can be attributed to these large site-specific biases, which may be caused by the high spatiotemporal variability of surface soil moisture, and the scale differences between the target point-scale soil moisture and 1-km model predicted soil moisture. Specifically, in random and year-independent validation strategies, part of the site-specific information is known to the models; whereas in the site-independent validation method, this information is entirely unknown to the model. By adopting the TC method, it is possible to select soil moisture stations that are representative of the average

soil moisture on a larger scale, thereby alleviating the scale difference issue to some extent. However, there may still be large biases between measurements from these point-scale representative soil moisture stations and footprint-scale average soil moisture values. As these biases are site-specific, can be positive or negative, and have a median value for all samples near 0, the overall ubRMSE that the model achieved on the site- or year-independent test samples can still be large when these biases are unknown to the model. Nevertheless, training the model with representative soil moisture stations not only improved the model's overall performance over unknown spatiotemporal locations (Table 4), but also improved the performance metrics of the model at each station (Fig. 5).

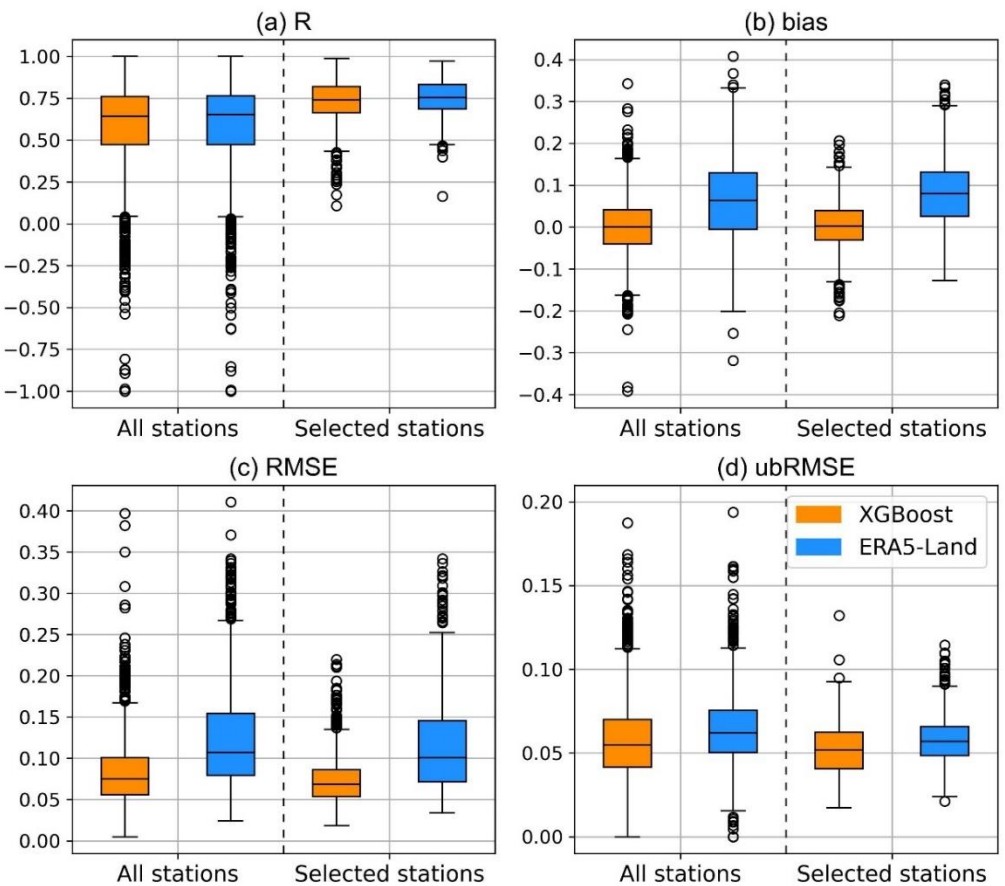

**Figure 5** Boxplots of the (a) R, (b) bias, (c) RMSE, and (d) ubRMSE achieved by the XGBoost models (blue) developed using all stations and the representative stations selected by the TC method, respectively, in comparison with those of the ERA5-Land soil moisture product (orange).

In addition to the performance metrics of the two XGBoost models at each station, Table 6 shows the validation accuracies of the model developed using the representative stations over different land cover types. Affected by a series of practical factors, the distribution of ISMN soil moisture stations is uneven in space, with the majority of the stations located in the CONUS. After screening stations via the TC method, the

spatial distribution of representative stations remained uneven, with the resulting number of stations for each land cover type also varying significantly (Fig. 1). Overall, the performance of the model developed using the representative stations for most land cover types showed an improvement compared with the model developed using all stations, as indicated by larger median R values, and smaller median RMSE and ubRMSE values. However, the median ubRMSE of the model achieved for forests was larger than that for other land cover types, likely a result of soil moisture maintaining at high levels in forested areas. Additionally, among the seven land cover types, the model achieved the lowest median R values for shrublands and barren lands, likely due to the limited number of stations present across these two types. However, the model also achieved the lowest median ubRMSE values for these two types, which can be partly attributed to the fact that despite the low sample percentages, the number of samples for these land cover types was sufficient for the models to learn, and in part due to the relatively small soil moisture dynamics of these two types. Although the median bias of the model for each land cover type was near 0, the model exhibited a large prediction bias for most stations across each land cover type (Table 6). After removing the prediction bias at each station, the median ubRMSE of the model for the seven land cover types ranged from 0.031 to 0.061 $m^3$ $m^{-3}$, marking a dramatic decrease over the corresponding median RMSE. Given that a large prediction bias existed in each land cover type, and that the model performance did not vary significantly across different types, it was suggested that the uneven distribution of land cover types across samples was not the major cause of the decreased overall model accuracy over unknown spaces.

**Table 6** Performance metric statistics for the XGBoost models developed using all stations and representative stations, and those achieved by the latter model over each land cover type.

| Types | Num | R | | | Bias ($m^3$ $m^{-3}$) | | | RMSE ($m^3$ $m^{-3}$) | | | ubRMSE ($m^3$ $m^{-3}$) | | |
|---|---|---|---|---|---|---|---|---|---|---|---|---|---|
| | | med | min | max | med | min | max | med | min | max | med | min | max |
| All stations | 1145 | 0.64 | -1.0 | 1.0 | 0.00 | -0.39 | 0.34 | 0.075 | 0.005 | 0.397 | 0.055 | 0.000 | 0.188 |
| Selected stations | 715 | 0.74 | 0.11 | 0.99 | 0.00 | -0.21 | 0.21 | 0.068 | 0.019 | 0.220 | 0.052 | 0.017 | 0.132 |
| Forests | 35 | 0.73 | 0.11 | 0.85 | 0.02 | -0.14 | 0.18 | 0.079 | 0.041 | 0.185 | 0.061 | 0.026 | 0.091 |
| Shrublands | 16 | 0.61 | 0.46 | 0.79 | -0.01 | -0.07 | 0.10 | 0.043 | 0.027 | 0.116 | 0.031 | 0.022 | 0.056 |
| Savannas | 185 | 0.77 | 0.24 | 0.97 | 0.01 | -0.17 | 0.18 | 0.070 | 0.019 | 0.194 | 0.051 | 0.017 | 0.132 |
| Grassland | 327 | 0.75 | 0.26 | 0.99 | 0.00 | -0.21 | 0.21 | 0.067 | 0.019 | 0.220 | 0.053 | 0.018 | 0.083 |
| Urban | 12 | 0.68 | 0.34 | 0.87 | 0.00 | -0.15 | 0.13 | 0.068 | 0.027 | 0.152 | 0.050 | 0.025 | 0.067 |
| Croplands | 130 | 0.73 | 0.29 | 0.89 | 0.00 | -0.20 | 0.21 | 0.065 | 0.030 | 0.214 | 0.049 | 0.026 | 0.106 |
| Barren | 10 | 0.57 | 0.27 | 0.82 | -0.03 | -0.07 | 0.08 | 0.050 | 0.028 | 0.090 | 0.034 | 0.025 | 0.056 |

**4.3 Validation of the GLASS SM product on independent networks**

Using the XGBoost model developed above, a global 1-km spatiotemporally continuous soil moisture

product (GLASS SM) was generated. To intuitively demonstrate the ability of this product for capturing the temporal variations in soil moisture over an unknown space, four independent networks under different climatic and environmental conditions were selected, and the time-series curves of the GLASS and measured soil moisture for these networks were compared. Considering the high spatiotemporal variability of surface soil moisture and the scale differences between point-scale observations and the 1-km GLASS SM product, the mean measured soil moisture curve was first calculated by averaging soil moisture curves from all stations within a network, and then compared with the mean predicted soil moisture curve calculated using all corresponding pixels of the GLASS SM product within that network. Moreover, as an input variable of the 1-km soil moisture estimation model, the time-series curves of the ERA5-Land reanalysis soil moisture product over the four independent networks were also extracted as a reference.

In most cases, the GLASS soil moisture curves were much closer to the measured values than the time-series curves of the ERA5-Land reanalysis soil moisture product in both the YA and YB soil moisture networks (Fig. 6 (a), (b)). The R values between the GLASS and measured soil moisture for these two networks were 0.84 and 0.89, respectively, which were slightly higher than the ERA5-Land soil moisture (0.80 and 0.84); whereas the ubRMSE values were 0.048 and 0.034 $m^3$ $m^{-3}$, respectively, slightly lower than the ERA5-Land soil moisture product (0.052 and 0.044 $m^3$ $m^{-3}$). Accordingly, over these two relatively dense soil moisture networks, the 1-km GLASS SM product can basically capture the dynamics of measured soil moisture. However, underestimates occurred at some high-value intervals on the measured soil moisture curves, which may be caused by nearby irrigation at some stations within agricultural regions, where the GLASS SM product may not be able to capture such patterns, given that irrigation is usually not uniformly distributed in space.

For the Fort Cobb and Little Washita soil moisture networks, both the GLASS and ERA5-Land soil moisture estimates basically captured the dynamics of measured soil moisture (Fig. 6 (c), (d)). Specifically, the R values between the mean GLASS and measured soil moisture for these two networks were 0.69 and 0.76, respectively, slightly lower than the ERA5-Land soil moisture product (0.74 and 0.77). However, both the GLASS and ERA5-Land reanalysis soil moisture products showed a large positive bias throughout most of the observation period, particularly in the Little Washita network. This is likely because these two soil moisture networks cover a relatively large watershed containing only a few stations. Nevertheless, the ubRMSE values between the mean GLASS and measured soil moisture values for these two networks were 0.037 and 0.033 $m^3$ $m^{-3}$, respectively, which were significantly lower than those for the ERA5-Land soil

moisture (0.047 and 0.046 m³ m⁻³). Overall, above results suggested that the derived product can accurately capture the temporal variations of in situ soil moisture under different climatic conditions. Further, the GLASS SM product achieved similar R values as the ERA5-Land product across these networks, with the R values ranging from 0.69 to 0.89 and ubRMSE values ranging from 0.033 to 0.048 m³ m⁻³.

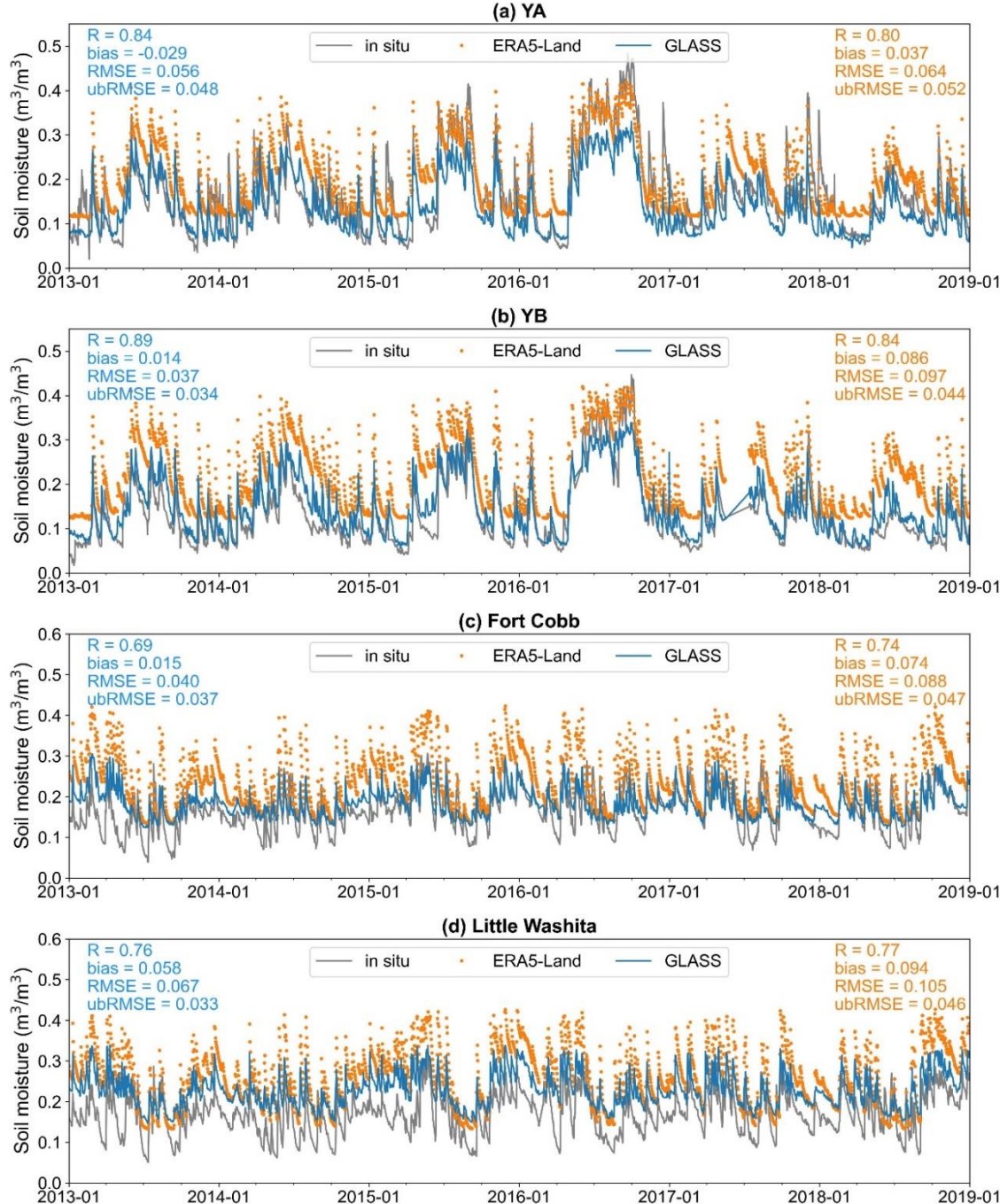

**Figure 6** Time-series plots of the mean in situ, ERA5-Land, and GLASS soil moisture for four independent soil moisture networks.

**4.4 Comparison with existing global soil moisture products**

After producing the global 1-km spatiotemporally continuous GLASS SM product, it was compared with

two global microwave soil moisture products for spatiotemporal consistency. The first product selected for comparison was SPL2SMAP_S, the first publicly released global soil moisture product at a spatial resolution of 1 km. Because the SPL2SMAP_S 1-km product has a temporal resolution of 12 days over most global areas and it has many spatial gaps at the daily scale, spatial synthesis of the SPL2SMAP_S was conducted during a 12-day period with relatively high spatial coverage before comparison. Figure 7 shows the spatial distribution of the SPL2SMAP_S 1-km soil moisture product, synthesized from 3 to 15 October 2016, alongside the 1-km spatiotemporally continuous GLASS SM map for 9 October 2016. Here, it can be seen that the 12-day synthetic SPL2SMAP_S soil moisture product still has large spatial gaps (e.g., the western continental United States, western China, and southwestern Australia); whereas the GLASS SM product has a substantially more complete spatial coverage (except for the high-latitude regions during the cold seasons). With regards to the spatial distribution characteristics, both soil moisture products with 1 km resolutions exhibits a high level of consistency, with higher soil moisture levels found in the tropics, eastern U.S., and southeastern China, and lower levels observed in deserts (e.g., Sahara) and other semi-arid regions.

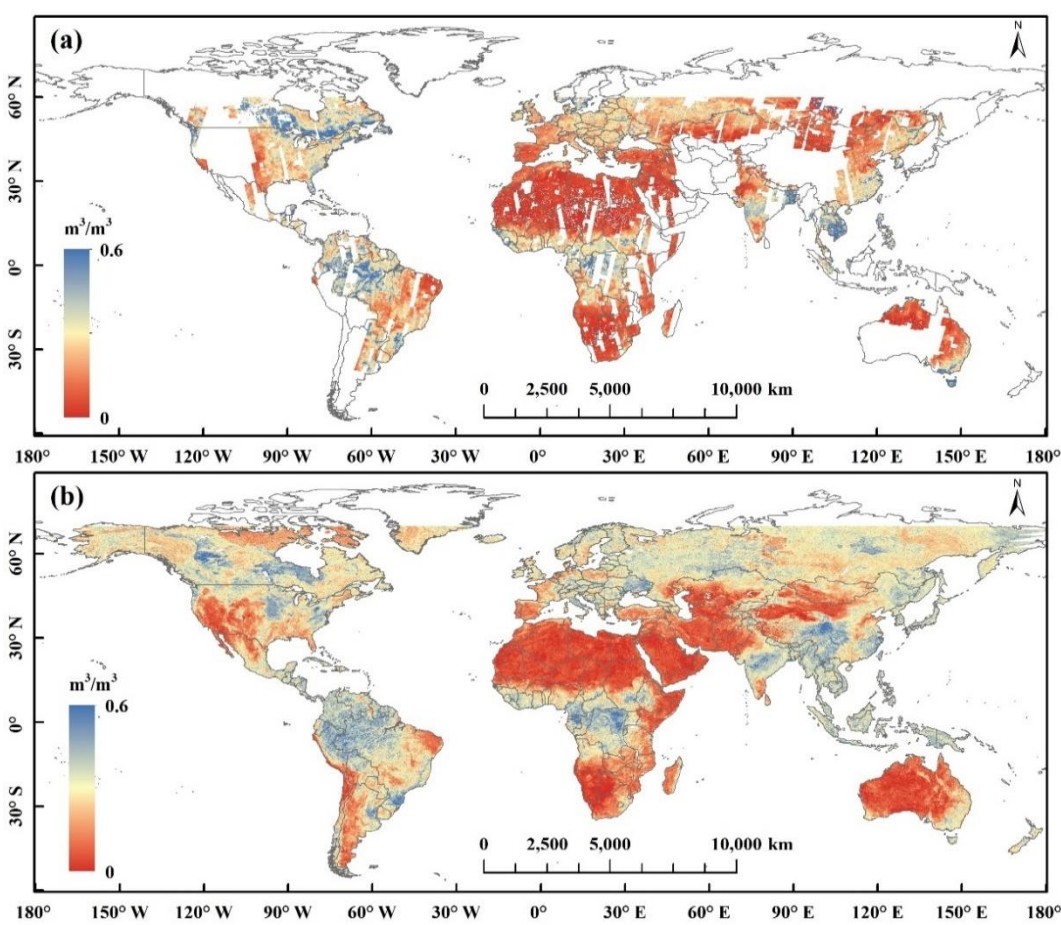

**Figure 7** (a) 12-day synthetic SPL2SMAP_S 1-km soil moisture map from 3 to 15 October 2016, and (b) the

1-km spatiotemporally continuous GLASS SM map on 9 October 2016.

To quantitatively investigate the spatial consistency between these two 1-km soil moisture products, spatial R and RMSD between them were calculated for each 12-day of 2016 using collocated pixels, after removing soil moisture estimates larger than 0.6 m³ m⁻³ from the SPL2SMAP_S product. As displayed in Fig. 8 (a), the spatial R (orange line) between the GLASS and SPL2SMAP_S products ranges from 0.61 to 0.67, with a median value of 0.62, partially affected by the discontinuous spatial coverage of the SPL2SMAP_S product. The spatial RMSD (orange dots) between the two 1-km products in 2016 ranges from 0.098 to 0.106 m³ m⁻³, and the relatively large RMSD values may be attributed to the greater spatial heterogeneity (e.g. terrain and soil texture) at fine scales which could cause large disparities in soil moisture estimates from different algorithms. Overall, both qualitative and quantitative comparisons suggested a good and stable spatial consistency between the 1-km GLASS and SPL2SMAP_S microwave soil moisture products.

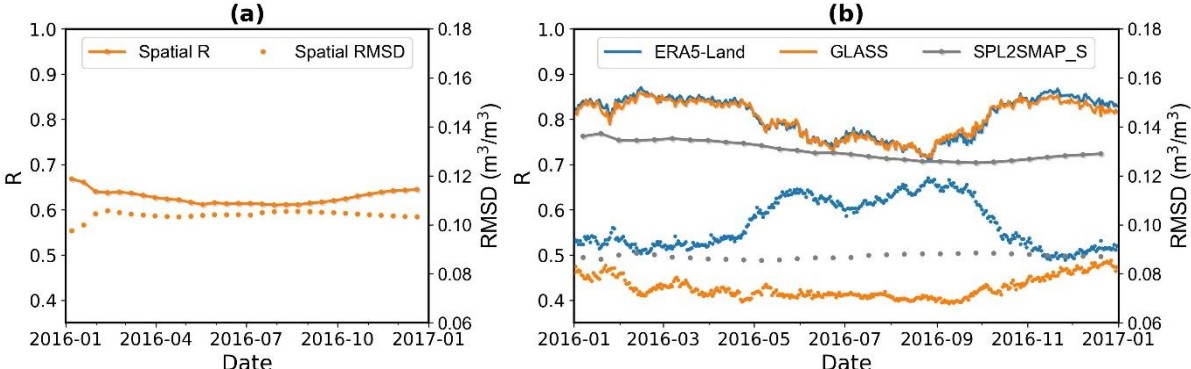

**Figure 8** Time-series plots of the spatial R (lines) and RMSD (dots) calculated between (a) the GLASS and SPL2SMAP_S soil moisture products at 1 km resolution and (b) the ESA CCI and three resampled soil moisture products (ERA5-Land, GLASS, and SPL2SMAP_S) at 0.25° resolution in 2016.

The second global product selected for comparison was the widely used ESA CCI combined soil moisture dataset with a spatial resolution of 0.25°. Because the CCI soil moisture product has a daily temporal resolution and more complete spatial coverage, more quantitative analyses can be conducted when comparing with the 1-km spatiotemporally continuous GLASS SM product. Figure 9 shows the spatial distribution of the CCI active–passive microwave combined soil moisture and GLASS SM resampled to 0.25° for four days from different seasons in 2016, as well as the corresponding scatterplots of these two soil moisture products. The high spatial consistency between the CCI soil moisture product and resampled GLASS SM product on different dates is readily apparent, as both products display lower soil moisture values in arid regions, including the western U.S., northern and southern Africa, Middle East, central and western Asia, and Austria,

and higher soil moisture values in tropical and temperate regions, such as central Africa, southern Asia, the eastern U.S., and southeastern China. Although CCI estimates incorporate a variety of active and passive microwave soil moisture products, its spatial coverage remains incomplete partly due to observation gaps of the sensors, and the physical limitations of microwave soil moisture retrieval algorithms (Dorigo et al., 2017), such as failing to provide accurate soil moisture predictions on densely vegetated land surfaces (e.g., the Amazon River and Congo basins). In contrast, the GLASS SM product shows greater spatial integrity, except at high latitudes in cold seasons due to low temperatures and frozen soils.

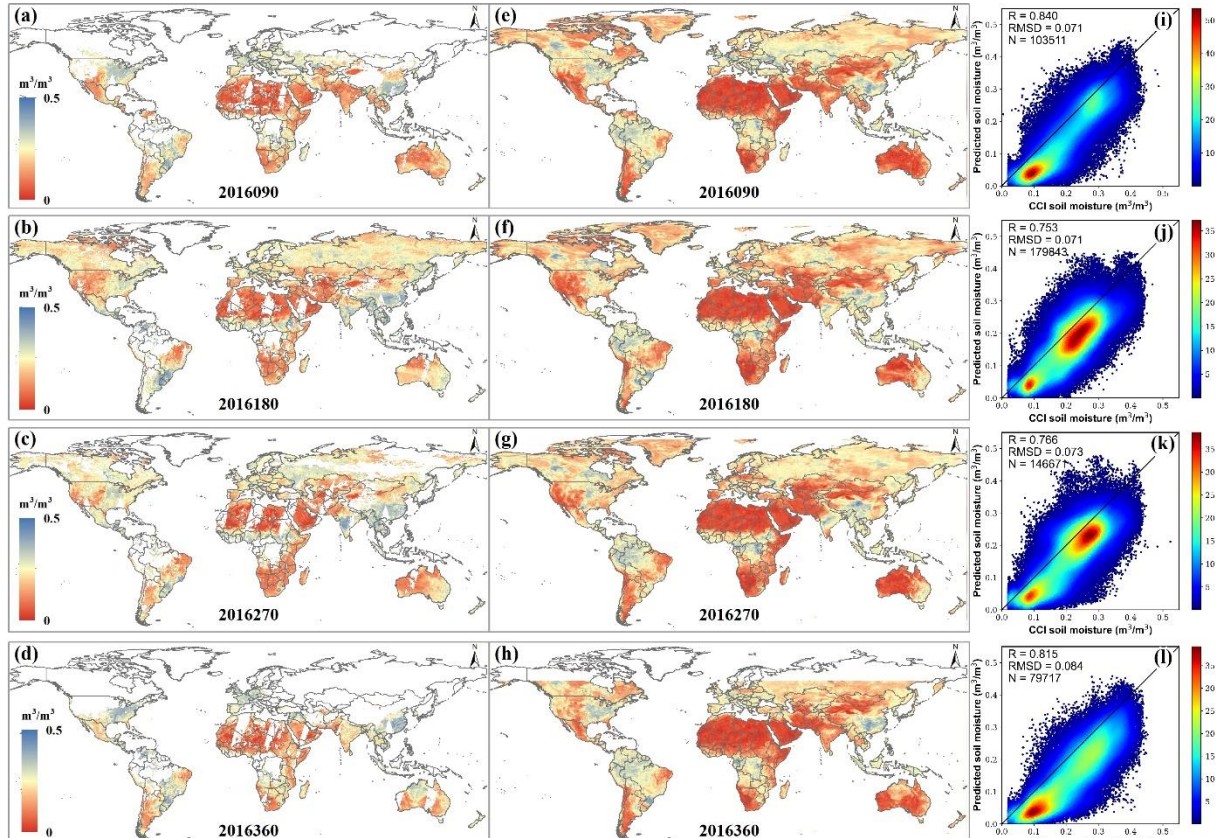

**Figure 9** (a–d) ESA CCI combined soil moisture maps at 0.25°, (e–h) the corresponding spatiotemporally continuous GLASS SM maps resampled to 0.25°, and (i–l) scatterplots of the two products for four Julian dates (90, 180, 270, 360) selected from different seasons of 2016.

As shown in Fig. 8 (b), the daily spatial R between the resampled GLASS and ESA CCI soil moisture products at 0.25° resolution in 2016 ranges from 0.72 to 0.86, with a median value of 0.82, indicating that the two products exhibit high spatial consistency across the seasons. As a comparison, the spatial R and RMSD between the CCI and two other resampled soil moisture products (ERA5-Land and SPL2SMAP_S) were also calculated and plotted. It's clear that the spatial R curves of the resampled ERA5-Land (blue) and GLASS (orange) at 0.25° only differ slightly, which is to be expected given that the coarse-scale ERA5-Land

soil moisture was used to provide background soil moisture information for our model and it achieved the highest importance score among all the input variables. Both curves exhibit significant seasonal variation, with higher spatial R values in spring and winter than in summer or autumn, possibly related to the larger differences between the two resampled products (GLASS and ERA5-Land) and CCI over high latitudes. However, the spatial RMSD curves of the ERA5-Land and GLASS differ significantly. While the blue dotted line (RMSD between CCI and ERA5-Land) exhibits an opposite seasonal pattern to the R curves, with RMSD ranging widely from 0.086 to 0.12 $m^3$ $m^{-3}$, the orange dotted line (RMSD between CCI and GLASS) is more stable, with RMSD ranging from 0.068 to 0.087 $m^3$ $m^{-3}$. Besides, as also shown in Fig. 8 (b), although the resampled SPL2SMAP_S soil moisture product has the most stable spatial R and RMSD curves (gray), it achieves relatively lower spatial R values and larger spatial RMSD values than those of the resampled GLASS product 0.25°, suggesting its relatively lower level of spatial consistency with the CCI product. This is to our surprise considering that both the SPL2SMAP_S and CCI soil moisture products were derived from microwave satellite observations, and a possible cause for this could be the discontinuous spatial coverage of the SPL2SMAP_S product.

Note that the GLASS SM product displays a general underestimation relative to the CCI combined soil moisture (Fig. 9 (i–l)). Although the overestimation of the CCI soil moisture product has been reported in previous study, particularly for Equatorial (Savanna) regions (Al-Yaari et al., 2019), the GLASS SM product may also contain some biases, which jointly contribute to the RMSD between them. Figure 10 shows a zoomed-in comparison between the 1-km GLASS and 0.25° ESA CCI microwave soil moisture product in western China on 28 June 2016, with the corresponding 0.1° ERA5-Land reanalysis soil moisture product, which is one of the main inputs to the XGBoost model, also shown as a reference. In general, the GLASS product exhibits spatial consistency with both coarse-scale soil moisture products, with lower soil moisture levels in the Junggar Basin, Tarim Basin, Qaidam Basin, and western part of the Tibetan Plateau, and higher soil moisture levels in the Tianshan Mountains, Ili River Valley, and southeastern part of the plateau where the vegetation is also much denser. Specifically, in the southeastern Tibetan Plateau, the GLASS and CCI soil moisture products show higher consistency, while the ERA5-Land soil moisture product is suspected to be underestimated. Moreover, it is clear that the 1-km GLASS SM product is not only spatially complete, but also contains more spatial details which can well reflect the distribution patterns of terrain and vegetation.

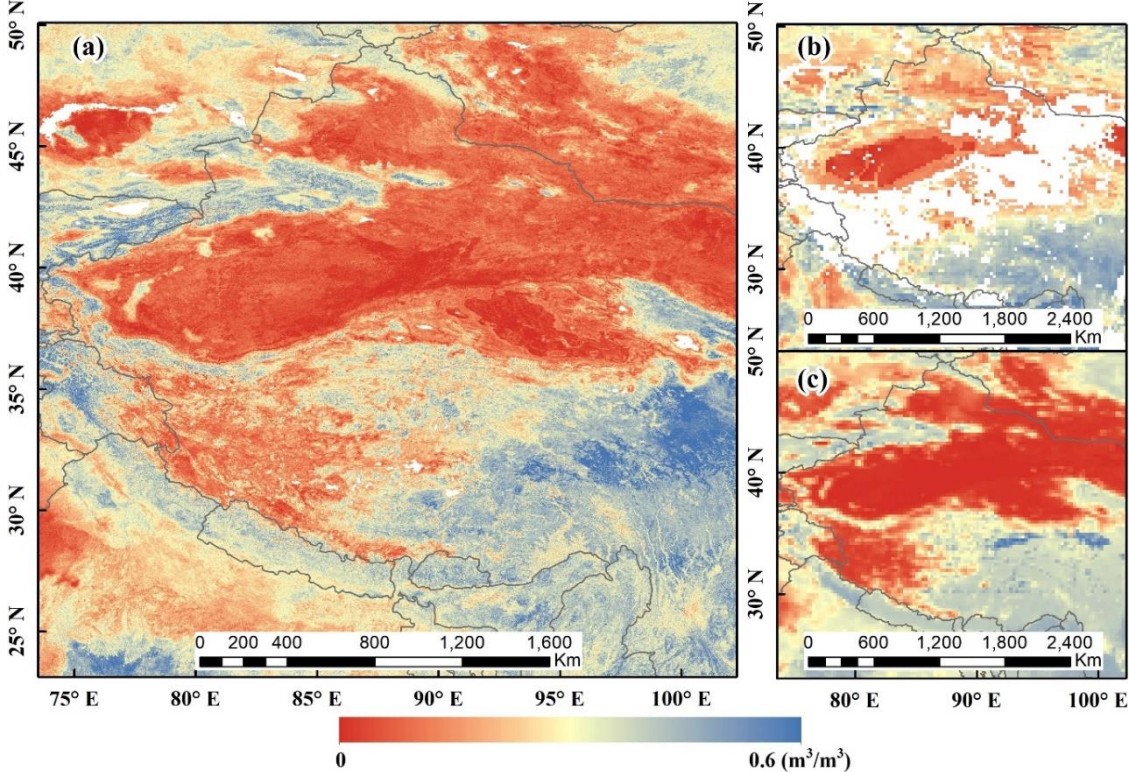

**Figure 10** Zoomed-in comparison of the (a) 1-km GLASS, (b) 0.25° ESA CCI, and (c) 0.1° ERA5-Land soil moisture products in western China on 28 June 2016 (the 180th day).

In addition to the spatial consistency analysis described above, the temporal consistency between the CCI and spatiotemporally continuous GLASS SM product was also explored. Specifically, for each pixel of these two products with > 30 days of concurrent predictions, the R and RMSD between the time-series soil moisture predictions were calculated separately for 2016, and the spatial distribution of these two metrics is shown in Fig. 11. The correlation between two products was high in most areas, except the Sahara Desert, high latitudes, and some localized regions. The relatively low or even negative R values between the two products in the Sahara Desert is likely due to that soil moisture in this region is close to zero, and a small difference in temporal variation may lead to poor correlation. It can also be seen from Fig. 11 (b) that the RMSD values between the two products in the Sahara Desert were rather small. The relatively low R values between the two products at high latitudes may be attributed to the irregular prediction frequency of the CCI product at high latitudes, and the rapid change in soil moisture during the freeze–thaw transition period in this region, which possibly cause larger errors in both products and thus increased temporal inconsistency. Greater differences between soil moisture products at high latitudes have also been found elsewhere (Wang et al., 2021). Further, no obvious patterns were revealed regarding the distribution of RMSD between the two soil moisture products, as the regions with relatively large RMSD values were rather scattered.

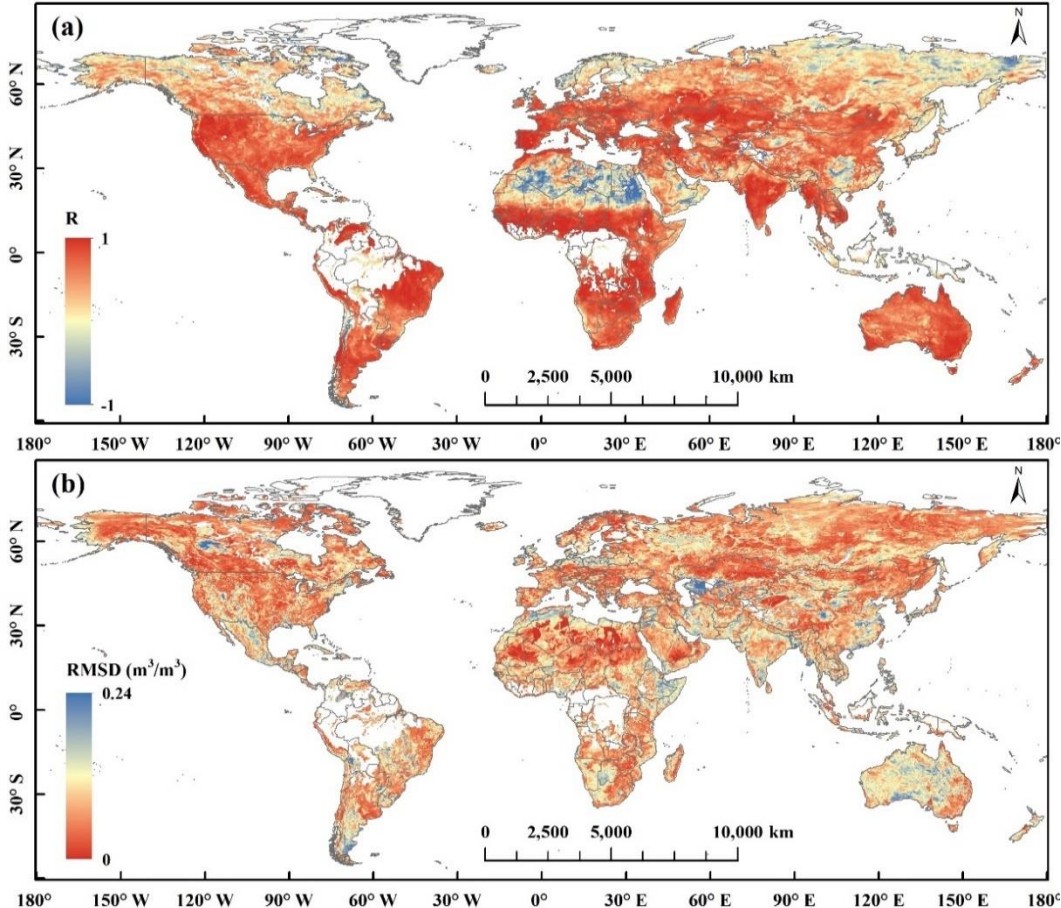

**Figure 11** The spatial distribution of (a) R and (b) RMSD between the ESA CCI combined soil moisture product and the spatiotemporally continuous GLASS SM product in 2016.

## 5 Discussion

To address the lack of high-resolution, spatiotemporally continuous global soil moisture products, this study developed a global 1-km soil moisture estimation framework which integrated multi-source datasets using an XGBoost model. This framework was adapted from the 30 m soil moisture estimation framework proposed by zhang et al. (2022b), in which the Landsat 8 surface reflectance and thermal observations were replaced with the spatiotemporally continuous GLASS albedo, LST, and LAI products, to mitigate the

influence of clouds on the spatial continuity and temporal resolution of soil moisture product. Meanwhile, the relatively high temporal resolution of GLASS products allows for much more collocated training samples, which are supposed to alleviate the underestimation of the original 30 m model at high soil moisture levels. In addition, considering the relatively large scale differences between point-scale in situ soil moisture datasets and GLASS products compared to Landsat datasets, the TC method was adopted to select the representative

soil moisture stations and their measurements were used as the training target of the model. Results showed

that the 1-km soil moisture estimation model achieved satisfactory overall accuracy and training the model with representative stations selected by the TC method can considerably improve its performance over unknown time and space.

Most of previous machine learning-based studies aimed at soil moisture estimation have divided the samples from all observation locations and times randomly into training and test datasets. In this case, model's accuracy on the random test samples may seem rather high as a result of model overfitting, because these test samples may not be spatially or temporally independent of those in the training dataset and part of the site-specific information is disclosed to the model. Therefore, model performance must also be fully evaluated using samples from unknown time or space. Senyurek et al.'s (2020) trained a random forest model using the Cyclone Global Navigation Satellite System observations, as well as the ISMN in situ soil moisture and other geophysical datasets, which was then fully evaluated using a 5-fold cross-validation, site-independent, and year-based techniques. Before the model training process, several critical screening conditions were applied to select 106 stations from the 234 ISMN soil moisture stations over the CONUS, and the 5-fold cross-validation R and RMSE of the random forest model were 0.89 and 0.052 $m^3$ $m^{-3}$, respectively; whereas the site-independent cross-validation R and RMSE values were 0.64 and 0.088 $m^3$ $m^{-3}$, respectively. Similarly, the overall R and RMSE of the 1-km GLASS SM model for the random and site-independent test samples were 0.941, 0.038 $m^3$ $m^{-3}$, and 0.715, 0.079 $m^3$ $m^{-3}$, respectively. Notably, Senyurek et al. (2020) attributed the relatively lower site-independent validation accuracy to the fact that different soil moisture stations have distinct climatology, which is difficult for the machine learning model to capture without bias. Instead, we argue that the high validation accuracy achieved by the machine learning models on the random test samples is mostly likely a result of overfitting, while the relatively lower site-independent validation accuracy is much more realistic. The authors further suggested that model performance could be improved by increasing the representativeness of various land surface conditions within training datasets. Although a representative training dataset is essential for data-driven machine learning models, it was found here that a large prediction bias existed across all land cover types and the resulting model performance did not vary significantly among them. Therefore, it was concluded here that the site-specific biases induced by scale differences rather than the uneven distribution of land cover types among samples are the major cause of the decreased overall accuracy of the model over unknown time and space.

As emphasized in Gruber et al. (2020), despite that downscaled soil moisture products usually provide more spatial details visually, they may not reflect real soil moisture variations, and it is thus necessary to

estimate the spatial R for the downscaled products, in addition to temporal analyses. Then, Crow et al. (2022) defined the success of a downscaling algorithm as either achieving better temporal accuracy or spatial skill than the original coarse-scale product that is interpolated onto the fine-scale spatial grid. As can be seen from Fig. 5 (a), the temporal R values achieved by the XGBoost model at representative stations are similar to those of the coarse-scale ERA5-Land soil moisture product, and Fig. 8 (b) shows that the GLASS and ERA5-Land products achieved similar Spatial R values when they are both resampled to 0.25° resolution. Therefore, to identify whether the 1-km GLASS SM product actually have added value with respect to the 0.1° ERA5-Land product, we also calculated the spatial R for the XGBoost model on a daily basis using soil moisture measurements from representative stations, and then compared it with that of the ERA5-Land product interpolated onto the 1 km grid. To make the comparison more rigorous, soil moisture estimated using the 5-fold cross-validation method from the model was adopted to calculated the spatial R, instead of the final GLASS SM product (yielding even better results). As displayed in Fig. 12, the spatial R values achieved by the XGBoost model at representative stations improve significantly compared to those of the ERA5-Land product, with the median spatial R increasing from 0.60 to 0.66, and in most cases, the difference in spatial R (R_diff) between the XGBoost model and ERA5-Land product is positive, with a median value of 0.06. Accordingly, it is reasonable to believe that the 1-km GLASS SM product does provide more spatial information which reflect fine-scale soil moisture variations, rather than just adding ineffective details.

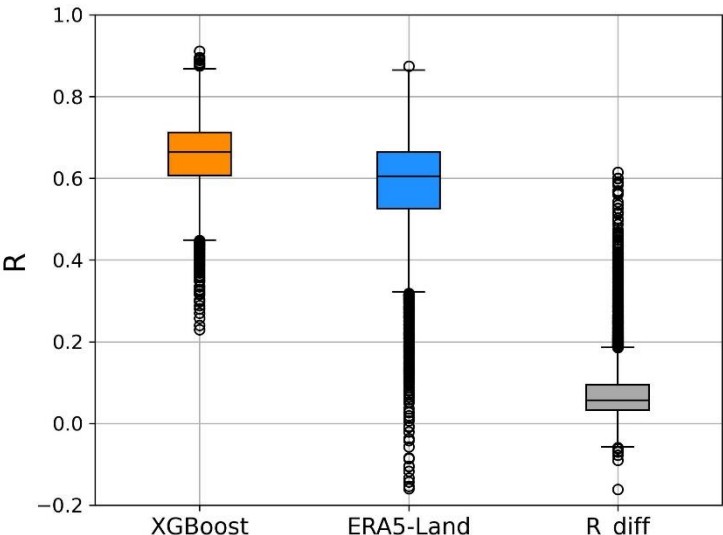

**Figure 12** Boxplot of the spatial R calculated for the XGBoost model on a daily basis using soil moisture measurements from representative stations, in comparison with those of the ERA5-Land product. The difference in spatial R between the XGBoost model and ERA5-Land product is denoted as R_diff.

To date, several studies have attempted to further improve the accuracy of machine learning based soil moisture estimation models through different strategies. Abbaszadeh et al. (2019) classified in situ soil moisture stations within the CONUS according to soil texture class, developing 12 distinct random forest models to downscale the SMAP 36-km soil moisture product using atmospheric, geophysical and in situ soil moisture datasets. Their downscaled 1-km soil moisture product achieved good overall validation accuracy on both core validation sites and 300 sparse soil moisture stations, with the proposed downscaling approach outperforming the uniform downscaling approach. Similarly, Karthikeyan and Mishra (2021) clustered CONUS into 11 homogeneous regions using a k-means algorithm based on a range of climate and landscape variables, before training an XGBoost model for each region and soil layer to downscale the SMAP Level 4 soil moisture product. Validation at 79 independent soil moisture stations showed that the downscaled product successfully captured temporal variations of measured soil moisture. We also have attempted to classify the ISMN stations based on their soil texture classes, or climatic and environmental properties prior to separately developing the models, however, the overall prediction accuracy did not seem to improve significantly.

Moreover, to mitigate the impacts of scale differences and improve the prediction accuracy, we also trained a distinct XGBoost model (Model 2) using the average soil moisture of all 30-m pixels within a 1-km pixel where the station was located as the target variable, which was calculated using the 30-m soil moisture estimation model developed by Zhang et al. (2022b). The overall accuracies of Model 2 and the previously developed model trained directly using in situ soil moisture (Model 1) on the YA and YB networks were then compared (Fig. 13). Here, it was found that Model 1 achieved good overall prediction accuracy for both networks; but as also shown in Fig. 6, Model 1 showed slight underestimation at higher soil moisture levels, especially in the YA region. In contrast, while Model 2 obtained similar R values as Model 1, it exhibited much more sever underestimation at higher soil moisture levels in both the YA and YB networks. This may be attributed to the lack of high soil moisture samples in the original 30 m soil moisture estimation model, which were even scarcer after averaging to 1 km. To further improve Model 2 accuracy, uniform global sampling can be performed to generate a large number of 1-km averaged soil moisture samples, but this would be rather labor intensive. Alternatively, the global 1-km GLASS SM product generated using Model 1 accurately captured the temporal variations of the in situ soil moisture, and exhibited high spatiotemporal consistency with microwave soil moisture products, although some site-specific biases may exist while validating the product against sparse soil moisture stations. Future studies could focus on mitigating the impacts of scale differences on the machine learning models, either by deploying more dense soil moisture

monitoring networks, or by further improving the accuracy of high resolution (e.g., 30 m) but often spatiotemporally discontinuous soil moisture products, and then training the 1-km spatiotemporally continuous GLASS SM model directly using the higher resolution soil moisture products.

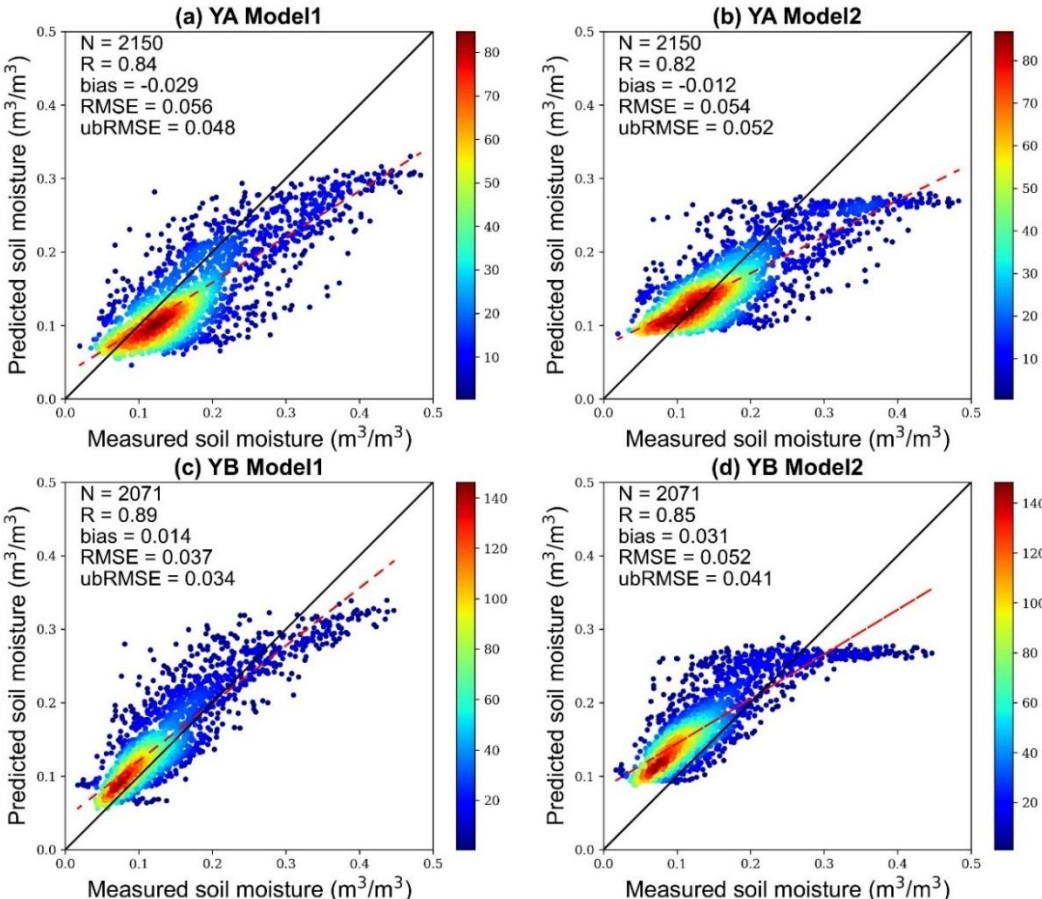

**Figure 13** Scatterplots of mean measured and predicted soil moisture from different models on the: (a–b) YA, and (c–d) YB soil moisture networks. Point colors indicate the probability density; whereas the red dashed line is the linear regression, and the black solid line is the 1:1 relationship.

## 6 Data availability

The global daily 1-km spatiotemporally continuous soil moisture product (GLASS SM) from 2000 to 2020 is freely available at http://glass.umd.edu/soil_moisture/. In addition, for user's convenience, the annual average global soil moisture dataset at 1 km resolution was also generated, which can be downloaded from https://doi.org/10.5281/zenodo.7172664 (Zhang et al., 2022a). Note that this product represents the volumetric water content in the uppermost soil layer (0–5 cm). Files are stored in the Sinusoidal projection and "GeoTIFF" format.

## 7 Conclusions

A global 1-km spatiotemporally continuous soil moisture product (GLASS SM) was derived here using an

XGBoost ensemble learning model that integrated multi-source datasets, including remotely sensed GLASS products, ERA5-Land reanalysis products, as well as ground-based ISMN soil moisture, and static auxiliary datasets. Validation of the XGBoost model was conducted using three complementary validation strategies and the GLASS SM product was also evaluated across four independent networks, demonstrating the product's strong capacity to capture temporal dynamics of measured soil moisture. This global 1-km soil moisture product also exhibited high spatiotemporal consistency with two global microwave soil moisture products. Overall, the main findings of the study can be summarized as follows:

(1) When the samples from all stations and years were randomly divided into training and test datasets, the XGBoost model achieved a high accuracy on the random test samples. By using the TC method to select representative stations, the validation accuracy of the model was further improved significantly, with an overall R and RMSE of 0.941 and 0.038 $m^3$ $m^{-3}$, respectively. Nevertheless, such high accuracy achieved by the model on the random test sample is clearly a result of overfitting,

(2) Training the model with representative stations selected by the TC method also considerably improved its performance for site- or year-independent samples (i.e., over unknown time and space). The overall validation accuracy of the model trained using representative stations on the site-independent test samples, which was least likely to be overfitted, was an R of 0.715 and RMSE of 0.079 $m^3$ $m^{-3}$. Compared to the model developed without station filtering, the accuracies of the model trained using representative stations improved significantly on most stations, with the median R and ubRMSE of the model for each station increasing from 0.64 to 0.74, and decreasing from 0.055 to 0.052 $m^3$ $m^{-3}$, respectively.

(3) The time-series validation results of the 1-km GLASS SM product over four independent networks indicated that the product can accurately capture temporal variations in measured soil moisture under different climatic conditions. The GLASS SM product achieved similar R values as the ERA5-Land product, with the R values ranging from 0.69 to 0.89 and ubRMSE values range from 0.033 to 0.048 $m^3$ $m^{-3}$.

(4) Compared with the 1-km SMAP/Sentinel-1 SPL2SMAP_S soil moisture product and the ESA CCI active–passive microwave combined soil moisture product at 0.25°, the global 1-km spatiotemporally continuous soil moisture product generated here had a more complete spatial coverage, and exhibited high spatiotemporal consistency with these two products.

The long-term (2000–2020) global GLASS SM product with high spatiotemporal resolution (1 km, daily) and reliable accuracy generated here can benefit climate change studies, hydrological modeling, and agricultural applications at regional and global scales. It is also a valuable complement to currently released

global microwave and model-simulated soil moisture datasets. Future studies could consider further improving and fully evaluating the accuracy of the GLASS SM product.

**Author contributions.** SL and YZ developed the methodology and designed the experiments. YZ, HM, BL, JX, GZ, XL, and CX collected and preprocessed the data. YZ carried out the experiments. YZ, TH, and QW produced the product. YZ prepared the manuscript with contributions from all co-authors.

**Competing interests.** The authors declare that they have no conflict of interest.

**Disclaimer.** Publisher's note: Copernicus Publications remains neutral with regard to jurisdictional claims in published maps and institutional affiliations.

**Acknowledgements.** We would like to thank the GLASS team for providing albedo, LST, and LAI products; the ECMWF project for offering the ERA5-Land soil moisture product; the SoilGrids project for the soil property dataset; and Yamazaki's team for the MERIT DEM. We also appreciate the scientists and networks who have shared their valuable ground-based soil moisture datasets, as well as the ISMN project for making these datasets readily accessible. We are also very grateful to the editors and reviewers for their valuable suggestions, which helped us a lot to improve the manuscript.

**Financial support.** This study was supported by the National Natural Science Foundation of China (grant no. 42090011).

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
