# Peer review of "Generation of global 1-km daily soil moisture product from 2000 to 2020 using ensemble learning"

_Earth System Science Data, 2022_

## Author Comment (AC1)

**CC1: 'Comment on essd-2022-348', Noemi Vergopolan**

Nice paper, and a very interesting effort. However, there is a critical issue with how the split sample was performed, that needs to be addressed to ensure the transparency and transferability of the method's accuracy results.

1. Given how in situ soil moisture data is collected (often clustered at specific basins), if you performed random or site-independent sampling, you are likely training and validating your results with sites that belong to the same basins (even if you perform K-fold cross-validation). This does not ensure your model performance is independent and can be transferred to other locations (or continents) where no training data is available.

2. Likewise, covariate input data for your trained model can cover grids of up to 25 km resolution (ESA CCI). This means that if you select evaluation sites within the 25 km grid space of a training site, you are evaluating your model with the same covariates used on a site to which the model was already trained, thus cross-contaminating your evaluation results.

Both issues with how the split sample is performed can inflate your performance results. To ensure the transferability of your accuracy results where no in-situ observations are available, all the split sample cases should ensure no evaluation sites are within (at least) 25 km distance from a training site. Otherwise, the accuracy of your approach is only representative of a few meters distances from a training site.

Response: We would like to thank Dr. Noemi Vergopolan for the constructive comment on our manuscript. We agree that the way of splitting sample is crucial for evaluating the model's accuracy when developing a machine learning model. Therefore, we adopted three complementary validation strategies to fully evaluate the model performance: random, site-independent, and year-independent. Besides, we adopted four independent soil moisture datasets, which were not included in the training dataset (ISMN soil moisture dataset), to assess the transferability of the model.

Among the four independent soil moisture networks, there are two micronets deployed in the two watersheds in southwestern Oklahoma, the Little Washita River watershed comprises 611 km$^2$ and the Fort Cobb Reservoir watershed comprises 813 km$^2$ (their locations are added to the Fig.1 in the revised manuscript). Since soil moisture measurements from these two watersheds were not used for model developing, the model's performance on them can reflect the its transferability in unknown basins (Sect. 4.3). Nevertheless, a fully validation of the GLASS SM product against other locations is rather necessary, if more in situ soil moisture datasets are accessible, apart from the ISMN database.

Regarding the second point you mentioned, firstly, we shall clarify that the ESA CCI

product was not used as a covariate input data for our model, but only used to select the representative ISMN stations. The lowest spatial resolution of the input datasets is that of the ERA5-Land soil moisture product (~ 10 km). Secondly, we carefully considered your suggestion and designed an experiment in which no validation sites were within 25 km distance from any training site and re-evaluated the model's accuracy. In the original site-independent validation process, four-fifths of the representative ISMN stations (572) were randomly selected to develop the model, and the remaining one-fifth stations (143) were used to assess the model accuracy, with the model achieving an R of 0.715 and RMSE of 0.079 m $^3$ m $^{-3}$. In the modified experiment, after removing some of the stations from the test dataset based on the suggested distance constraint (< 25 km), there were 71 stations left and the model achieved an R of 0.745 and RMSE of 0.078 m $^3$ m $^{-3}$. As there is no significant difference in accuracy, we believe that the original site-independent validation method did not inflate the model's performance.

**RC1: 'Comment on essd-2022-348', Anonymous Referee #1**

**General comments:**

The paper presents a framework for estimating SM at 1 km resolution combining reanalysis SM (ERA5-Land, ~10km) and other variables using the XGBoost algorithm. I find it an interesting approach and certainly there is an increasing interest from the hydrological community to have similar products from various applications.

Thank you for your positive and constructive review comments, we have revised the manuscript accordingly.

However, to make it even more robust, I would suggest the authors to further elaborate on the following points:

- The comparison with the SMAP/S1 product is very interesting and, in principle, of high value. However, I believe that a more thorough assessment would be beneficial, as currently there is no quantitative comparison and the analysis only takes into account one single day. The authors could, for instance, calculate spatial correlation among the 2 datasets for each 12-days overlapping samples and plot the corresponding time series.

Response: We have added the quantitative comparison between the GLASS and SPL2SMAP_S products as you suggested. Time-series plot of the spatial R and RMSD between the two products for each 12-day of 2016 is displayed in Fig. (8a). (Line 575–582)

- Similar to the previous point, I think it would be valuable to present a time series of spatial correlation between the GLASS SM and the ESA CCI products – in addition to the analysis presented by the authors.

Response: We have also added the time-series plot of the spatial R and RMSD between the 0.25° ESA CCI and resampled GLASS soil moisture products (Fig. (8b)). As a comparison, the spatial R and RMSD between the CCI and resampled ERA5-Land soil moisture products were also calculated and plotted. The analysis of this plot is given in Line 607–617.

- As ERA5-Land SM is the dataset being downscaled, it should be included a comparison of the GLASS and ERA5-Land SM products (as advocated in Gruber et al., 2021; Crow et al., 2022). Do we see a degradation in terms of temporal metrics (correlation, ubRMSD, ..) when looking at GLASS SM compared to ERA5-Land? Or does it improve even the temporal dynamics?

Response: We have added the performance metrics of the ERA5-Land soil moisture product at each station to Fig. 5 for comparison. (Line 460–475)

It was found that, at most of the representative stations, our model obtained similar or even larger R values compared to the ERA5-Land soil moisture product. However, there were also several stations where the model achieved relatively lower R values, yet this degradation in temporal metrics with respect to the original coarse-scale products can be found in many soil moisture downscaling studies (Gruber et al., 2020).

- line 83-85: there is also an operational SM product from Sentinel-1 generated for Europe (https://land.copernicus.eu/global/products/ssm, Bauer-Marschallinger et al., 2019)

Response: Thank you for the information, we have added the reference to this product into Line 81–83.

- Table 1 is not exhaustive; for instance, CGLS SSM and the EUMETSAT H-SAF SM (https://hsaf.meteoam.it/Products/ProductsList?type=soil_moisture) products are missing

Response: We have added these two soil moisture products and some other products to make Table 1 more complete. (Line 120)

- line 176: there is a more recent paper about the ISMN (Dorigo et al., 2021)

Response: Thanks for the reminder, we have replaced the ISMN paper to the more recent version. (Line 175)

- Section 2.6: here the revisit time of the SMAP/S1 product is not specified. Please add.

Response: We have added the temporal resolution of the SPL2SMAP_S product to Section 2.6. (Line 209)

- Figure 2: I would use a different color for the training stations – purple might be not so effective depending on the background color. Also, it would be interesting to know the distribution with respect to climate classes; maybe the authors could add a table with % of land cover and climate classes

Response: Thanks for the suggestion. We have changed the color for the training stations into red which shall make them more conspicuous. The locations of the four independent soil moisture networks used for validation were also added to Fig. 1.

   In addition, we added a table (Table 3) to show the number and percentage of training stations for each climate class and land cover type. (Line 179–189)

- Table 3 indicates that accounting for spatial variability is more important than accounting for the temporal component, as found in previous studies (e.g. Zappa et al., 2019).

Response: Thank you for the valuable comment. We have learned this paper and added this statement into the revised manuscript. (Line 388–391)

- Figure 4: for better readability and comparison of feature importance, I would combine the 3 plots into a single one with different colors for the 3 test experiments.

Response: Thanks for your advice. But the permutation feature importance ranking results for each model are different, and when they are combined into a single plot, the importance ranking (which feature is more important for a model) would be disrupted. Besides, the permutation feature importance scores are calculated relative to the accuracy of a certain model, thus the absolute importance scores of different models are not comparable. Therefore, we decided not to combine the three plots, yet we set

different colors for each group of input datasets to improve the readability of this figure. (Line 422)

- Table 4: rename the input datasets, eg coarse SM (ERA5-Land), vegetation and LST (GLASS), terrain, soil texture.
Response: We have renamed the input datasets in Table 5 as you suggested. (Line 448)

- Figure 5: create different subplots for the different metrics, as the spread of values on the y axis is rather wide.
Response: We have made separate subplots for different metrics as you advised. In addition, the performance metrics achieved by the ERA5-Land soil moisture product at each station are also added to Fig. 5 for comparison. (Line 490)

- Figure 9: invert or change the colorbar, as it might be a bit misleading at first sight to see that high correlation (good) is shown in red, but also high RMSD (poor result) has the same color.
Response: This figure has been revised accordingly. (Line 647)

Gruber, A., De Lannoy, G., Albergel, C., Al-Yaari, A., Brocca, L., Calvet, J.-C., Colliander, A., Cosh, M., Crow, W., Dorigo, W., Draper, C., Hirschi, M., Kerr, Y., Konings, A., Lahoz, W., McColl, K., Montzka, C., Muñoz-Sabater, J., Peng, J., Reichle, R., Richaume, P., Rüdiger, C., Scanlon, T., van der Schalie, R., Wigneron, J.-P., and Wagner, W.: Validation practices for satellite soil moisture retrievals: What are (the) errors?, Remote Sens. Environ., 244, 111806, https://doi.org/https://doi.org/10.1016/j.rse.2020.111806, 2020.

**RC2: 'Comment on essd-2022-348', Anonymous Referee #2**

This paper presents a gap free 1 km global soil moisture data set from 2000 - 2020 created using a XGBoost model. Such a data set is certainly relevant for the community and the manuscript perfectly fits the scope of ESSD. The setup of using XGBoost on coarse-resolution model data together with different high-resolution variables is certainly innovative and a sound attempt to improve spatial information content over other existing products. I also very much appreciate the effort the authors have put into clear language and a well structured manuscript.

I only have a few comments regarding clarity and the validation approach, which I hope the authors will consider to revise the manuscript:

Thank you for the valuable and constructive comments, which helped us to improve the quality of our manuscript.

**General comments:**

- Correlations of soil moisture and in situ data above 0.8-0.9 are completely unrealistic and strongly suggest over-fitting. The authors certainly tried to avoid overfitting using different strategies (year- and site-independent sampling), but I think the manuscript can benefit from a more explicit discussion on the issue of over-fitting, XGBoost's proneness to it, and when you do and do not suspect it to occur in your particular study.

Response: Thanks for your suggestion. Based on the validation results, it seems that the good performance of the XGBoost model on the random or year-independent test samples is clearly a result of model overfitting, while the relatively lower accuracy achieved by the model on site-independent test samples is least likely to be overfitted and can be regarded as the model's true accuracy. We have added this point to the Results section (Sect. 4.2) (Line 384–388) and emphasized it in both the Abstract and Conclusions.

- In general, references are appropriate, but often come a bit late. For example, triple collocation and XGBoost are already discussed in the introduction, but references to these methods only two sections later. I recommend referencing always upon first occurence. Along the same lines, I would provide the map of used ISMN locations already in the data sets section where I expected it.

Response: We have added the references for triple collocation and XGBoost in the introduction (Line 110, 114), and the position of the ISMN location map has also been adjusted.

- Soil moisture behaves very differently at point scale and at a 1 km scale, that is, they are in fact expected to exhibit different means and standard deviations just because of the scale difference. Training on point data somewhat imposes these point-scale properties on the data, so evaluating the model results in terms of bias, RMSD, and ubRMSD makes little to no sense, because comparing these metrics e.g., with those attained by ERA5 doesn't really say anything about which of these is actually more accurate, at the 1 km scale. This should be at least discussed.

Response: Thank you for the constructive comment. We agree that due to the scale difference, comparing the bias of our model with those attained by ERA5-Land, and claiming that our model can effectively reduce the large bias contained in the reanalysis soil moisture product doesn't make much sense. Thank you for pointing this out, and we have removed such statements from the manuscript. Still, we decide to keep the four metrics, because we can compare the R and ubRMSE between different models or products and the bias and RMSE are kept as reference. (Line 332–334)

- When considering the case of least potential overfitting (i.e., the site-independent validation), evaluation results, at least in terms of temporal correlation (which are actually the relevant ones; see above) are VERY close to those attained by ERA5, which begs the question of why bothering with the sophisticated ML in the first place, and not use ERA5 directly? Perhaps better discuss the actual benefits of your product.
Response: The reason of using the ML model was to obtain a high-resolution (1 km) global soil moisture product by integrating multi-source auxiliary products at higher spatial resolution with the coarse-scale (~9 km) ERA5-Land soil moisture product. We have added a zoomed-in comparison plot between the GLASS and ERA5-Land SM products to show that the 1-km GLASS product contains much richer spatial details. (Line 620–631)

- To me, the distinction between Sec. 4 and 5 isn't clear. It feels like Sec 5 is just a continuation with more results. Also, why hasn't the use of two different model setups not been discussed in the methods section? That comes a bit out of the blue.
Response: Thank you for the advice. We didn't put Model 2 into the Method section because we didn't want to overcomplicate the description of the algorithm. We chose to describe Model 2 in the Discussion section because we also tried several other methods from the literature to improve the model accuracy in this section. We have slightly modified the text to make the intention of this experiment clearer. (Line 703)

**Specific comments:**
- L65--: It's a strong claim that all global products have poor accuracy over densely vegetated areas, which I wouldn't consider to be true (considering that "densely vegetated" draws too vague a line). I recommend to delete this sentence
Response: Thank you for the advice, we agree that this is a strong claim and have deleted this sentence.

- L71 I suspect "combining" should be "using" or something similar? "combining", in that sentence, doesn't make sense.
Response: The sentence has been revised to make it clearer. (Line 69–71)

-L70-85: Somewhat a personal preference, but I believe this paragraph is largely a repetition of what is already discussed at length in Peng et al. (2020). For brevity, I'd remove it and just refer to this paper.
Response: Thanks for the advice. If I'm not mistaken, you should be referring to Peng

et al. (2021). We have added this paper for the reader's reference (Line 67). But this paper mainly focuses on the applications of high-resolution satellite soil moisture products, while we aimed to list the global or regional soil moisture products at fine scales developed using different data sources and algorithms in recent years. Besides, about half of the literature does not appear in Peng et al. (2021). Therefore, we decided to keep this paragraph in the manuscript.

L145: Blank missing before "study" and the opening bracket.
Response: Corrected.

-L149: Typo: EAR5 (same in Fig. 6)
Response: Thank you for the reminder, we have corrected this mistake.

-L177: Old link. Should be https://ismn.earth/
Response: We have changed the link to the new version. (Line 176)

Sec 2.5: Aren't these data sets also in the ISMN? Also, it'd be nice if Fig 2 would have a different color coding for training stations and these independent networks.
Response: We have checked and confirmed that these independent soil moisture datasets are not in the ISMN database. Note that although the OZNET is included in the ISMN dataset, the two dense subnetworks (YA and YB) are currently not included.
    We have added the locations of the four independent soil moisture networks used for validation to Fig.1 and set a different color for them as you suggested. (Line 185)

L211: Insufficient referencing for the CCI product; see https://esa-soilmoisture-cci.org/node/236
Response: Thank you for pointing this out. We have completed the references for the CCI product. (Line 221)

Sec. 3.2: Perhaps add references to previous work that has used TC as an indicator for spatial representativeness of in situ stations.
Response: We have added the references that used the TC method to analysis coarse-scale spatial representativeness of in situ soil moisture data to Sect. 3.2. (Line 259–261)

Sec. 3.3: Perhaps list the used hyperparameters
Response: Thanks for the advice. We have added the key hyperparameters set for the model to the manuscript. (Line 303–307)

L337: I didn't understand the difference between "representative stations selected using TC" and "stations excluded using TC".
Response: We have added an explanation after "the stations excluded using the TC method": "not included in the representative stations". (Line 348)

- L425-430: I am not convinced that your validation approach proves that your product actually contains more spatial detail. As mentioned in the general comments, you mainly show that biases relative to point scale measurements are smaller, but a 1 km product SHOULD be biased with respect to that scale; showing only that biases are smaller than those with ERA5 doesn't exclude the possibility that the ML isn't over-doing the bias correction.

Response: As mentioned above, we have removed such statements from the manuscript. A zoomed-in comparison plot between the GLASS and ERA5-Land SM products was also added to show that the 1-km GLASS product contains much richer spatial details. We hope that our revisions have addressed your concerns. (Line 620–631)

Fig. 7; Why not showing any metrics for the SMAP / GLASS SM comparison?

Response: We have added a time-series plot of the spatial R and RMSD between the two products for each 12-day of 2016, as shown in Fig. 8 (a), to provide a quantitative comparison between the GLASS and SPL2SMAP_S SM products. (Line 575–582)

Fig. 9: Why not comparing ERA5 also to the CCI product, or SMAP?

Response: We have added a time-series plot of the spatial R and RMSD between the CCI and ERA5-Land SM products for each day of 2016, as displayed in Fig. 8 (b) (Line 607–617). In addition, we have also added a zoomed-in comparison plot between these two products, as shown in Fig. 10. (Line 630)

- L626: Arguably, the "relatively lower site-independent validation accuracy" is MUCH more realistic. I would rather attribute the higher accuracy in the other cases to over-fitting.

Response: Thank you very much for your constructive comments, we have added this point to the manuscript. (Line 681–683)

- L665: It's a circular argument that "future studies should focus on improving the accuracy of soil moisture products at higher resolution" to then train your model to create more accurate soil moisture products at high resolution.

Response: Here, we mean that training the model using high resolution but often discontinuous soil moisture products. We have modified this sentence to make it clearer. (Line 720–722)

- L687: This R value is clearly a result of overfitting.

Response: We have emphasized this in the conclusions. (Line 746–747)

- L699: As discussed above, reducing biases is not necessarily a meaningful goal when comparing 1 km products to point scale products.

Response: We have removed this statement from the manuscript as you suggested.

- L705: remove "the"

Response: Corrected.

Peng, J., Albergel, C., Balenzano, A., Brocca, L., Cartus, O., Cosh, M. H., Crow, W. T., Dabrowska-Zielinska, K., Dadson, S., Davidson, M. W. J., de Rosnay, P., Dorigo, W., Gruber, A., Hagemann, S., Hirschi, M., Kerr, Y. H., Lovergine, F., Mahecha, M. D., Marzahn, P., Mattia, F., Musial, J. P., Preuschmann, S., Reichle, R. H., Satalino, G., Silgram, M., van Bodegom, P. M., Verhoest, N. E. C., Wagner, W., Walker, J. P., Wegmüller, U., and Loew, A.: A roadmap for high-resolution satellite soil moisture applications – confronting product characteristics with user requirements, Remote Sens. Environ., 252, 112162, https://doi.org/10.1016/J.RSE.2020.112162, 2021.

**RC3: 'Comment on essd-2022-348', Anonymous Referee #3**

**General comments:**

This paper presents a study of high-resolution seamless soil moisture estimation based on the machine learning model and multi-source datasets, which can fill the gap of the lack of long-term global soil moisture products with both high spatial and temporal resolutions and benefit a range of applications. The subject is of interest to the scientific community and the description is clear.

Three complementary model validation strategies (random, site-independent, and year-independent) were adopted by the authors, which avoids over-fitting to a large extent and makes the model more robust. I found this manuscript is well structured and suggest for acceptance after addressing the following concerns, partially related to the models and validation of the soil moisture data obtained.

Thank you for the positive review and valuable suggestions. We gratefully appreciate your help with this manuscript.

**Specific comments:**

1. Why did you choose the XGBoost model? Have you ever tried other machine learning models (Random Forest) or deep learning models (e.g. LSTM) besides XGBoost?

Response: The XGBoost model is chosen because it is simple, stable and fast. We also tried the widely used RF model, while it can achieve similar accuracy as the XGBoost model, it is much slower than the XGBoost model during both training and predicting processes. For example, it takes half an hour for the trained RF model to predict the 1-km global soil moisture product for one day, while the XGBoost model only takes about 10 minutes. On the other hand, although deep learning models are characterized by improved performance and higher flexibility, they require significant computational resources and large amounts of samples to train effectively. As a limited number of in situ soil moisture data were used as the training target of our model, if the LSTM model were used, time-series samples would have to be created and the number of samples available for training the model would be even smaller. Therefore, after considering the complexity, accuracy and speed of the models, the XGBoost model is chosen.

2. Although the ISMN data are used as input, in most areas where there is no ground observation, the EAR5-Land soil moisture reanalysis data is the main input factor of this algorithm. The importance of the ERA5-Land data is extremely high. Does this mean that the accuracy of the ERA5-Land data itself is one of the most important factors affecting the accuracy of the developed data set? If so, the author should explain it in the text. At the same time, in view of the extremely high weight of ERA5-Land, does it mean that it is the basis or background value of the proposed 1-km data, while other parameters are only corrected in a local range (0.1°*0.1°)? If so, how is this method fundamentally different from other machine learning-based soil moisture downscaling methods?

Response: The accuracy of the ERA5-Land product did affect the accuracy of the developed data set. Over most of the representative stations, the XGBoost model obtained similar or even larger R values compared to the ERA5-Land soil moisture product. However, there were also several stations where the model achieved relatively lower R values, yet this degradation in temporal metrics with respect to the original coarse-scale products can be found in many soil moisture downscaling studies. We have added this point into the manuscript. (Line 469–473)

ERA5-Land soil moisture product was chosen as one of the inputs to the model because it can provide reliable background soil moisture information. But we don't think that other high-resolution input datasets only provide the local correction. As shown in Fig. 10. There is a notable difference between the GLASS and ERA5- ERA5-Land products in the southeastern part of Tibetan Plateau. (Line 630)

Machine learning models are data-driven and the models are similar. The difference mainly lies in the selection of input datasets and the training process of the models. We used the TC method to select the representative stations to address the scale differences issue. Three validation strategies (random, site-independent, and year-independent) were adopted to prevent the model evaluation process from overfitting. Fully evaluation and inter-comparison were also performed to ensure to accuracy of the product.

3. While comparing the coarse scale global soil moisture product with the 1-km GLASS soil moisture product (Section 4.4), in addition to evaluating the spatiotemporal consistency between them, the authors are suggested to add a zoomed-in plot to demonstrate the superiority of the GLASS product in terms of spatial resolution.
Response: We have added a zoomed-in comparison plot (Fig. 10) between the GLASS and two coarse-scale SM products, the ESA CCI and ERA5-Land SM products, to show that the 1-km GLASS product contains much richer spatial details. (Line 620–628)

4. To make Table 1 more comprehensive, the other soil moisture products can also be added to this table, for example a recent publicly released global product which was generated using multi-channel collaborative algorithm:
https://doi.org/10.11888/Terre.tpdc.272907
https://doi.org/10.11888/Terre.tpdc.272088
There are also others such as the SMOS 1-km soil moisture data and the Copernicus soil moisture data from Sentinel-1 (https://land.copernicus.eu/global/products/ssm) cover Europe and a similar study at: https://doi.org/10.11888/RemoteSen.tpdc.272760
Response: Thank you for the information, we have added above products into Table 1.

1. The location of independent in situ validation datasets (section 2.5) is also suggested to add into Figure 2.
Response: We have added the locations of the four independent soil moisture networks used for validation to Fig. 1 as you suggested. (Line 185)

2. What I don't quite understand is that the importance of the factors obtained through the random test, year-independent test, and site-independent are not consistent (Fig. 4),

indicating that the models trained by the three methods are different, which set of models is finally used for the final soil moisture data set production? Which should be explained in the manuscript.

Response: The final model was developed using all the representative ISMN stations, its feature importance results over unknown regions could refer to those calculated on the site-independent test samples. We have emphasized this point int the manuscript. (Line 419–421)

3. In the verification part, the author used the data from four ground observation networks to show that compared with the ERA5 data (Fig. 6), the overall bias and RMSE of the obtained 1-km soil moisture are smaller, which is a gratifying conclusion. However, the type of land cover represented by these networks are extremely limited, so the reviewers believe that this verification might not sufficient enough.

Response: Thank you for the suggestion. Fig. 6 is mainly intended to show the temporal consistency between the GLASS and in situ soil moisture on several typical dense soil moisture networks. The validation accuracy of the model over different land cover types can be referred to Table 6 (Line 518), which was calculated on the site-independent test samples (least likely to be overfitted) and could be regarded as the model's true accuracy over unknown spaces.

4. How does the RMSD used in Figure 9 differ from the RMSE metric should be clarified in Section 3.4.

Response: We have added the explanation of the root mean square difference (RMSD) at the end of Sect. 3.4 as you suggested. (Line 330–332)

5. In Table 5, please explain why "urban" is not excluded from the soil moisture data. Moreover, the difficulty of estimating soil moisture in forests and barren is greater than that of grasslands and croplands. However, in Table 5, there is not much difference in the accuracy of different types of results. Why? Perhaps the author can add a column of "Number of Sites" in Table 5, so that readers can better understand the accuracy of the model in various land cover types.

Response: We have added the number of sites in Table 6 for readers' reference (Line 518). The accuracies achieved by the model over forests (similar R but high ubRMSE) and barren (low R but low ubRMSE) are slightly lower than those over grasslands and croplands. There is not much difference in the accuracy of different land cover types may be because despite the limited number of stations for some land cover types, the number of samples for these types is sufficient. To facilitate management, several soil moisture stations are installed in the urban or build-up lands. After adopting the TC method, most of the urban type stations were screened out, leaving the more representative ones (with a relatively small percentage of impervious surface), so we decided to keep this type of soil moisture stations.

---

## Author Response (AR2)

Dear Dr. Zhao and Dr. Zhou:

Thank you very much for your time and effort in editing and reviewing our manuscript. We would also like to thank Dr. Vergopolan and three anonymous referees for their positive and constructive review comments. The manuscript has greatly benefited from these insightful suggestions.

We have carefully considered the review report returned by Anonymous Referee #2 and made some major revisions. First, we clarified the intention of Fig. 8 in the manuscript. Second, we added the spatial R and RMSD curves of the resampled SPL2SMAP_S product to Fig.8b as suggested. Third, we added a figure (Fig. 12) in the discussion section to demonstrate that the 1-km GLASS SM product have added value with respect to the 0.1° ERA5-Land product.

We hope that our revisions have addressed your concerns. Please find the point-by-point reply to the comments below.

Sincerely,
Yufang Zhang & co-authors

**Report #1**

I consider most of my comments properly addressed. I only have two comments left, which I hope the authors could address:

Thank you for your attentive review and insightful comments, we have carefully considered your suggestions and revised our manuscript accordingly.

- I find the newly added Figure 8 confusing.

8b shows that there is virtually no difference between GLASS (1km) vs. CCI (0.25 deg) and ERA5 (0.1 deg) and CCI (0.25 deg), which essentially suggests that there is no added spatial information in the 1km GLASS product than there is in the 0.1 degree ERA5 product. 8a compares GLASS with a 3km product, but that's not put in relation with the other products, so how much GLASS and SPL2SMAP_S agree isn't very informative in any regard. Wouldn't it make more sense to compare both GLASS and the SPL2SMAP_S to both ERA5 and CCI? Either way, please clarify the intention of the figure and which conclusion is drawn from it. (I don't find the RMSD comparison meaningful in any way, because all that says is that GLASS holds less bias, which is trivial because - as mentioned - bias is not a meaningful metric to compare SM products at different scale.

Response: Fig. 8 intended to quantitatively demonstrate the spatial consistency between the GLASS soil moisture product and two global microwave soil moisture products at

different spatial resolutions (1 km and 0.25°). We have clarified the intention of this figure in the manuscript. (Line 575–584)

It is true that the spatial R curves of the resampled ERA5-Land (blue) and GLASS (orange) at 0.25° only differ slightly, suggesting that the two products are rather close at the 0.25° spatial resolution. But this doesn't mean that the original 1-km GLASS product doesn't contain more spatial information than the 0.1° ERA5-Land product. We have added a figure in the discussion section to clarify this point, which would also be explained in detail in the response below.

Fig.8a aims to compares the GLASS and SPL2SMAP_S soil moisture products at 1 km resolution, and since the other soil moisture products (ERA5-Land and CCI) are of much lower resolution, they are not added to this plot. To make the comparison more informative, we have calculated the spatial R between the resampled SPL2SMAP_S and CCI products as you suggested and added the curve to Fig.8b. We found that it achieves relatively lower spatial R values than those of the resampled GLASS product, and a possible cause for this could be the discontinuous spatial coverage of the SPL2SMAP_S product. (Line 609–628)

We agree that while using point-scale in situ soil moisture dataset to validate soil moisture products at different resolutions, bias is not so meaningful. However, in Fig.8, we compare soil moisture products at the same resolution (1 km) or resampled to the same resolution (0.25°), and we believe that the RMSD between them can be severed as a reference, thus we decide to keep this metric in Fig.8.

- Spatial detail is confused with spatial information.

It is true that Figure 10a looks more detailed than Figs. 10b and c. However, one could also create a nice looking, highly-resolved plot at any resolution using randomly created data, but that doesn't mean that spatial variations would reflect true soil moisture variations in any way.

The authors acknowledge that the details "well reflect the distribution patterns of terrain and vegetation". Analyses regarding correlations w.r.t in situ data, especially spatial correlations, suggest that there isn't any more spatial INFORMATION in the GLASS data than there is in ERA5-Land, even though there is more spatial DETAIL. This is expected, of course, because the prediction model relies heavily on the ERA5 data, but it needs to be articulated accordingly. Pointing out that GLASS maps contain more "spatial detail" isn't useful, because the data suggest that, if anything, these spatial details do NOT reflect higher-resolution soil moisture variation.

Response: It can be seen from Fig.10 that, in the southeastern Tibetan Plateau where the vegetation is much denser, the GLASS soil moisture map shows higher soil moisture levels as expected, and it also exhibits consistency with the CCI product, while the

ERA5-Land product here is suspected to be underestimated. Therefore, we believe that the spatial details contained in the GLASS product can reflect soil moisture variations in a relatively reasonable way. (Line 638–641)

Nevertheless, we admit that Fig. 10 alone can't quantitatively demonstrate that the 1-km GLASS SM product have added value with respect to the 0.1° ERA5-Land product. Therefore, we calculated the spatial R for the XGBoost model on a daily basis using soil moisture measurements from representative stations, and then compared it with that of the ERA5-Land product interpolated onto the 1 km grid. Although the earlier results show that the temporal R values achieved by the XGBoost model at representative stations are similar to those of the coarse-scale ERA5-Land soil moisture product (Fig. 5 (a)), the spatial R values achieved by the XGBoost model improved significantly compared to those of the ERA5-Land product, with the median spatial R increasing from 0.60 to 0.66 (Fig. 12). Accordingly, it is reasonable to believe that the 1-km GLASS SM product does provide more spatial information which reflect fine-scale soil moisture variations, rather than just adding ineffective details. (Line 704–726)